# Differential impact of BTK active site inhibitors on the conformational state of full-length BTK

Raji E Joseph[1†], Neha Amatya[1], D Bruce Fulton[1], John R Engen[2], Thomas E Wales[2†*], Amy Andreotti[1*]

[1]Roy J. Carver Department of Biochemistry, Biophysics and Molecular Biology, Iowa State University, Ames, United States; [2]Department of Chemistry and Chemical Biology, Northeastern University, Boston, United States

**Abstract** Bruton's tyrosine kinase (BTK) is targeted in the treatment of B-cell disorders including leukemias and lymphomas. Currently approved BTK inhibitors, including Ibrutinib, a first-in-class covalent inhibitor of BTK, bind directly to the kinase active site. While effective at blocking the catalytic activity of BTK, consequences of drug binding on the global conformation of full-length BTK are unknown. Here, we uncover a range of conformational effects in full-length BTK induced by a panel of active site inhibitors, including large-scale shifts in the conformational equilibria of the regulatory domains. Additionally, we find that a remote Ibrutinib resistance mutation, T316A in the BTK SH2 domain, drives spurious BTK activity by destabilizing the compact autoinhibitory conformation of full-length BTK, shifting the conformational ensemble away from the autoinhibited form. Future development of BTK inhibitors will need to consider long-range allosteric consequences of inhibitor binding, including the emerging application of these BTK inhibitors in treating COVID-19.

*For correspondence:
T.Wales@northeastern.edu (TEW);
amyand@iastate.edu (AA)

†These authors contributed equally to this work

Competing interests: The authors declare that no competing interests exist.

## Introduction

Bruton's tyrosine kinase (BTK) is a non-receptor tyrosine kinase belonging to the TEC family (*Rawlings and Witte, 1995*; *Berg et al., 2005*). BTK is expressed primarily in B cells and myeloid cells, where it functions downstream of receptors including the B-cell receptor, Fc receptors, and toll-like receptors (*Kurosaki, 2011*; *López-Herrera et al., 2014*; *Xu et al., 2012*; *Hartkamp et al., 2015*; *Ellmeier et al., 2011*). B-cell receptor engagement leads to BTK activation (via phosphorylation on Y551 in the activation loop), which results in the direct phosphorylation and activation of phospholipase C gamma 2 (PLCγ2) triggering calcium flux and subsequent regulation of gene transcription (*Kurosaki, 2011*). In addition to the clear role of the BTK catalytic domain in activating PLCγ2, the N-terminal regulatory domains of BTK, which include the Pleckstrin homology and Tec homology (PHTH) domain, a proline-rich region (PRR), a SRC homology 3 (SH3) domain, and a SRC homology 2 (SH2) domain (*Figure 1a*), mediate autoinhibitory and activating interactions that control BTK function. Indeed, both kinase-dependent and independent activities of BTK have been shown to contribute to proper signaling downstream of immune receptors (*Saito et al., 2003*; *Halcomb et al., 2007*). Previous work has established that the regulatory SH3 and SH2 domains of BTK assemble into a compact autoinhibitory conformation on the 'distal' surface of the kinase domain, similar to the autoinhibited SRC kinases (*Figure 1b*; *Joseph et al., 2017*; *Wang et al., 2015*). Solution data support an additional autoinhibitory interface on the activation loop face of the kinase domain occupied by the PHTH domain (*Figure 1b*; *Amatya et al., 2019*; *Devkota et al., 2017*). Upon receptor-mediated activation, the regulatory domains of BTK bind their activating ligands, which releases the catalytic domain from its intramolecular interactions, thereby permitting

**eLife digest** Treatments for blood cancers, such as leukemia and lymphoma, rely heavily on chemotherapy, using drugs that target a vulnerable aspect of the cancer cells. B-cells, a type of white blood cell that produces antibodies, require a protein called Bruton's tyrosine kinase, or BTK for short, to survive. The drug ibrutinib (Imbruvica) is used to treat B-cell cancers by blocking BTK. The BTK protein consists of several regions. One of them, known as the kinase domain, is responsible for its activity as an enzyme (which allows it to modify other proteins by adding a 'tag' known as a phosphate group). The other regions of BTK, known as regulatory modules, control this activity. In BTK's inactive form, the regulatory modules attach to the kinase domain, blocking the regulatory modules from interacting with other proteins. When BTK is activated, it changes its conformation so the regulatory regions detach and become available for interactions with other proteins, at the same time exposing the active kinase domain.

Ibrutinib and other BTK drugs in development bind to the kinase domain to block its activity. However, it is not known how this binding affects the regulatory modules. Previous efforts to study how drugs bind to BTK have used a version of the protein that only had the kinase domain, instead of the full-length protein.

Now, Joseph et al. have studied full-length BTK and how it binds to five different drugs. The results reveal that ibrutinib and another drug called dasatinib both indirectly disrupt the normal position of the regulatory domains pushing BTK toward a conformation that resembles the activated state. By contrast, the three other compounds studied do not affect the inactive structure. Joseph et al. also examined a mutation in BTK that confers resistance against ibrutinib. This mutation increases the activity of BTK by disrupting the inactive structure, leading to B cells surviving better.

Understanding how drug resistance mechanisms can work will lead to better drug treatment strategies for cancer. BTK is also a target in other diseases such as allergies or asthma and even COVID-19. If interactions between partner proteins and the regulatory domain are important in these diseases, then they may be better treated with drugs that maintain the regulatory modules in their inactive state. This research will help to design drugs that are better able to control BTK activity.

activation loop phosphorylation and a conformational shift of the catalytic machinery from the inactive to active state.

BTK has become an important pharmacological target (*Bond et al., 2019*; *Feng et al., 2019*; *Kim, 2019*) due to its essential role downstream of immune receptors; for example, increased receptor signaling increases BTK activity driving multiple B-cell malignancies, autoimmune diseases, and allergies (*Pal Singh et al., 2018*; *Smith, 2017*; *Mohamed et al., 2009*). Ibrutinib is an irreversible ATP-competitive inhibitor that binds covalently to BTK C481 within the kinase active site (*Figure 1c, d*). This small molecule is the first BTK-specific inhibitor to be approved by the FDA and has become the frontline drug in efforts to control a variety of B-cell disorders. Crystallographic analysis reveals that Ibrutinib-bound BTK kinase domain adopts an inactive conformation (*Bender et al., 2017*); the activation loop is collapsed into the active site, the αC-helix is out, the regulatory spine is not assembled, and the conserved Lys/Glu salt bridge is broken (*Figure 1d*). Ibrutinib is currently approved for the treatment of chronic lymphocytic leukemia (CLL), Mantle cell lymphoma (MCL), Waldenstrom's macroglobulinemia, Marginal zone lymphoma (MZL) ,and chronic graft versus host disease and is in various stages of clinical trials for the treatment of other B-cell disorders (*Bond et al., 2019*; *Zi et al., 2019*; *Davids and Brown, 2014*; *Molica et al., 2020*; *Treon et al., 2015*; *Aw and Brown, 2017*; *Wang et al., 2013*). More selective, second-generation BTK inhibitors, acalabrutinib and zanubrutinib (*Barf et al., 2017*; *Guo et al., 2019*; *Wu et al., 2016*), have since received FDA approval bringing the total number of approved therapies that target BTK to three. Recent findings demonstrate a possible role for the BTK inhibitor acalabrutinib in treating excessive inflammation in the context of severe COVID-19 (*Roschewski et al., 2020*). Off-label use of acalabrutinib for hospitalized COVID-19 patients showed normalization of inflammatory markers and decreased oxygen requirements (*Roschewski et al., 2020*). Development of reversible BTK inhibitors has also

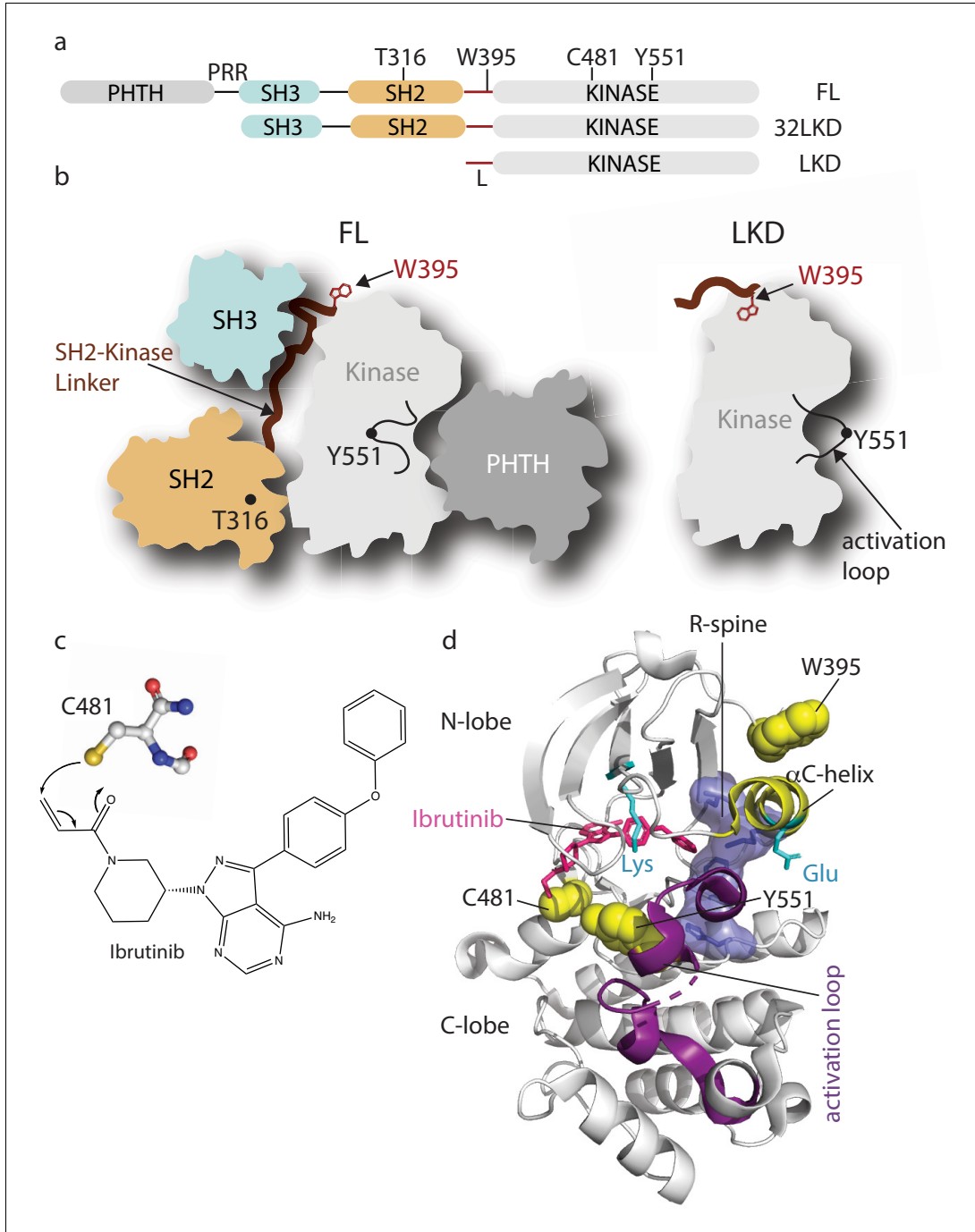

**Figure 1.** Domain organization of full-length Bruton's tyrosine kinase (BTK) and Ibrutinib binding site. (**a**) The BTK fragments used in this study are shown with key residues indicated above each domain. PHTH, Pleckstrin homology-Tec homology domain; PRR, proline-rich region; SH3, Src homology three domain; SH2, Src homology two domain; L, SH2-kinase linker; FL, full-length; 32LKD, SH3-SH2-linker-kinase domain; and LKD, linker-kinase domain. (**b**) Domain arrangement of BTK full-length (FL) in the autoinhibited conformation (left) based on the crystal structure of the BTK SH3-SH2-kinase (32LKD) fragment (PDB ID: 4XI2) and solution data supporting an autoinhibitory interaction between the PHTH domain and the activation loop face of the kinase domain (*Amatya et al., 2019*; *Devkota et al., 2017*). The location of T316 in the SH2 domain is indicated with a black circle. The active LKD fragment is depicted on the right. (**c**) The thiol group of C481 in the BTK kinase domain forms a covalent bond with the α,β unsaturated ketone of Ibrutinib. (**d**) Crystal structure (PDB ID: 5P9J) of BTK kinase domain (gray cartoon) bound to Ibrutinib, represented in pink sticks. The kinase domain N- and C-lobes are labeled, the activation loop is purple, the αC-helix yellow, side chains of the Lys/Glu pair are shown in

*Figure 1 continued on next page*

*Figure 1 continued*

cyan, sidechains of W395, Y551, and C481 are shown in yellow spheres, and the regulatory spine (R-spine) residues are shown as blue sticks and transparent surface.

progressed with a number of candidates in clinical development (*Brown et al., 2016*). To date, all BTK inhibitors currently being used or in clinal trials are active site inhibitors.

Despite the success of Ibrutinib in the clinic, drug resistance is a major ongoing challenge. Based on previous kinase-targeted therapies, it has become clear that acquisition of drug resistance occurs via a variety of mechanisms including loss of drug binding, activation of the kinase, or the activation of downstream substrates (*Lovly and Shaw, 2014*; *Sherbenou et al., 2010*; *Yun et al., 2008*). Of the Ibrutinib resistance mutations that have been identified in BTK, all except one (T316A) cluster to the BTK active site (*Puła et al., 2019*). The most prevalent mutation occurs at BTK C481, the residue in the kinase active site that covalently binds to Ibrutinib (*Figure 1d*; *Lampson and Brown, 2018*). The BTK C481S mutation and other kinase active site Ibrutinib resistance mutations are thought to interfere with stable Ibrutinib binding (*Furman et al., 2014*; *Woyach et al., 2014*). In contrast, the BTK T316A mutation occurs in the BTK SH2 domain (*Figure 1b*) and is the only Ibrutinib resistance mutation that has been identified to date outside of the BTK kinase domain (*Sharma et al., 2016*; *Kadri et al., 2017*). The mechanism by which the remote T316A mutation confers Ibrutinib resistance is currently unknown.

Although an abundance of data including pharmacology, associated toxicities, metabolic processing, clearance, potency, and efficacy are available for the full-length BTK protein (*Bond et al., 2019*; *Kim, 2019*; *Molica et al., 2020*; *Owen et al., 2019*), the major molecular-level insights provided by high-resolution BTK:drug complex crystal structures are limited to the kinase domain fragment of BTK with drug bound to the active site cavity (*Kuglstatter et al., 2011*; *Xing and Huang, 2014*). Structural consequences of drug binding to the full-length BTK protein have remained unexplored. Importantly, work from other kinase systems have shown that drug binding to the kinase active site can block catalytic activity as expected but can also have unanticipated and important long-range allosteric effects that impact protein/ligand interactions (*Sonti et al., 2018*; *Leonard et al., 2014*).

To expand our mechanistic understanding of drug resistance and the effect of different inhibitors on the full-length BTK protein, we developed solution-based Nuclear Magnetic Resonance (NMR) and Hydrogen Deuterium Exchange Mass Spectrometry (HDX-MS) tools to probe local and long-range conformational consequences of resistance mutation and drug occupancy in the BTK active site. A panel of five BTK inhibitors, Ibrutinib, GDC-0853 (Fenebrutinib), CGI1746, CC-292 (Spebrutinib, AVL-292), and Dasatinib, was assembled based on availability of crystallographic data and commercial availability of each drug. Additionally, drug choices were prioritized to represent a range of kinase structures (activation loop 'in' versus 'out', αC-helix 'in' versus 'out') and different inhibitor binding modes (covalent versus non-covalent). All inhibitors in this study, except Dasatinib, have been developed as BTK-specific inhibitors.

Our findings provide the first molecular insights into the conformational response of full-length BTK toward a spectrum of small-molecule active-site inhibitors. The inhibitors fall into two categories, differentiated by the presence or absence of an inhibitor-induced effect on the regulatory regions distant from the active site. We find that Ibrutinib and Dasatinib have profound effects on the entire BTK protein that extend beyond the active site. In contrast, GDC-0853, CGI1746, and CC-292 do not alter the conformational state of the regulatory domains. Additionally, we explore the mechanism of action of the T316A resistance mutation. This single remote amino acid mutation destabilizes the autoinhibited conformation of full-length BTK. We propose that an increase in the population of active enzyme contributes to the decreased Ibrutinib sensitivity observed in vivo. Taken together, these data provide a framework for developing new strategies to target BTK in the face of acquired drug resistance and reveal significant differences in the overall conformational preferences of full-length BTK bound to distinct small molecule inhibitors. We anticipate that further development of BTK inhibitors will rely on such information to fully understand, and ultimately control, the consequences of drug interactions within the full-length BTK protein.

## Results

### Ibrutinib binding to BTK kinase domain shifts the conformational equilibrium to the inactive conformation

A conserved structural switch in the αC-helix of protein kinases plays a crucial role in controlling catalytic activity (*Taylor et al., 2015*). The conformational equilibrium between an inactive and active kinase domain involves a switch in the αC-helix between an 'αC-out' and 'αC-in' conformation, respectively, which in turn either breaks or creates a salt bridge between a conserved Lys/Glu pair (*Figure 2a*). Crystal structures of active and inactive BTK kinase domain show that the conformational shift in αC is accompanied by a change in the rotamer conformation of the BTK W395 side-chain (*Figure 2a*). We have established use of the BTK W395 side chain indole NH resonance to monitor the conformational preference of the BTK kinase domain by solution NMR methods (*Joseph et al., 2017*). The upfield W395 resonance was previously assigned (*Joseph et al., 2017*) to the inactive, αC-out conformation and in the full-length apo BTK protein this resonance is in slow exchange with the downfield resonance corresponding to the active, αC-in kinase domain conformation (*Figure 2a*). This W395 resonance signature is consistent with previously established resonance frequencies of tryptophan indoles that are hydrogen bonded to water (*Fenwick et al., 2018*). The ~0.2 ppm upfield shift observed for the W395 indole NH proton likely reflects weaker hydrogen bonding in the more nonpolar environment of the W395 indole NH in the αC-out conformation compared to αC-in rather than a complete removal of the bound water, which is characterized by a > 2 ppm upfield shift (*Fenwick et al., 2018*).

To test whether Ibrutinib binding to the BTK kinase domain in solution stabilizes the αC-out conformation observed in the co-crystal structure (*Bender et al., 2017*), we monitored Ibrutinib binding to the linker-kinase domain fragment of BTK (*Figure 1a*, LKD, residues 381–659) focusing on the tryptophan indole region of $^{1}$H-$^{15}$N TROSY HSQC spectra (*Figure 2b*). The W395 indole NH resonance in the apo BTK LKD is in the downfield position (*Figure 2b*, top spectrum) showing that the LKD fragment, in the absence of Ibrutinib, adopts the active or αC-in conformation in solution consistent with activity assay results reported previously (*Joseph et al., 2007*). Titration of Ibrutinib into BTK LKD shows an increase in the intensity of the W395 peak in the upfield position, and a corresponding decrease in the intensity of the W395 peak in the downfield position (*Figure 2b*). This is consistent with a conformational change in solution from the αC-in to the αC-out state observed in the crystal structure of the kinase:Ibrutinib complex (*Figure 2b*). In fact, at a 1:1 molar ratio of protein to drug, the BTK LKD almost exclusively populates the inactive conformation consistent with a single binding site for Ibrutinib (*Figure 2b*). That the upfield resonance corresponds to W395 in Ibrutinib-bound BTK LKD was confirmed by acquisition of the same spectrum of the BTK W395A mutant bound to Ibrutinib (*Figure 2—figure supplement 1*). The intact mass analysis of the full-length BTK: Ibrutinib sample also supports the single drug binding site (*Figure 2c*).

Next, we acquired $^{1}$H-$^{15}$N TROSY HSQC spectra for full-length apo BTK and full-length BTK bound to Ibrutinib (*Figure 2d,iii*, *iv*). As previously reported (*Joseph et al., 2017*), full-length apo BTK (as compared to the BTK LKD) gives rise to more than one W395 resonance indicating exchange between αC-in and αC-out kinase domain conformations likely reflecting the increased conformational heterogeneity in the full-length protein due to the presence of the regulatory domains (*Figure 2d,iii*). Ibrutinib binding to full-length BTK shifts the conformational equilibrium to the αC-out conformation as observed for Ibrutinib-bound LKD (*Figure 2d*, *iv*). The greater W395 upfield shift induced by Ibrutinib compared to apo full-length BTK (red versus gray dashed line) may be due to local chemical shift changes in the presence of Ibrutinib and/or further stabilization of the αC-out conformer.

Ibrutinib binding to full-length BTK also results in appearance of a new peak corresponding to the indole NH resonance of a different tryptophan, W251 in the SH3 domain (*Figure 2d*, *iv*, peak denoted with asterisk). This resonance is broadened beyond detection in the apo full-length BTK spectrum due to intermediate exchange and so its appearance upon addition of Ibrutinib suggests that dynamics in the SH3 domain region have been altered. We have previously found that the W251 indole resonance is also visible in spectra of BTK Y223A (*Joseph et al., 2017*), a mutation that disrupts the autoinhibited conformation of BTK shifting the SH3 domain toward an unengaged state more similar to the isolated SH3 domain (*Figure 2—figure supplement 2*). Thus, the appearance of the W251 resonance in the spectrum of Ibrutinib-bound full-length BTK (*Figure 2d*, *iv*) suggests that

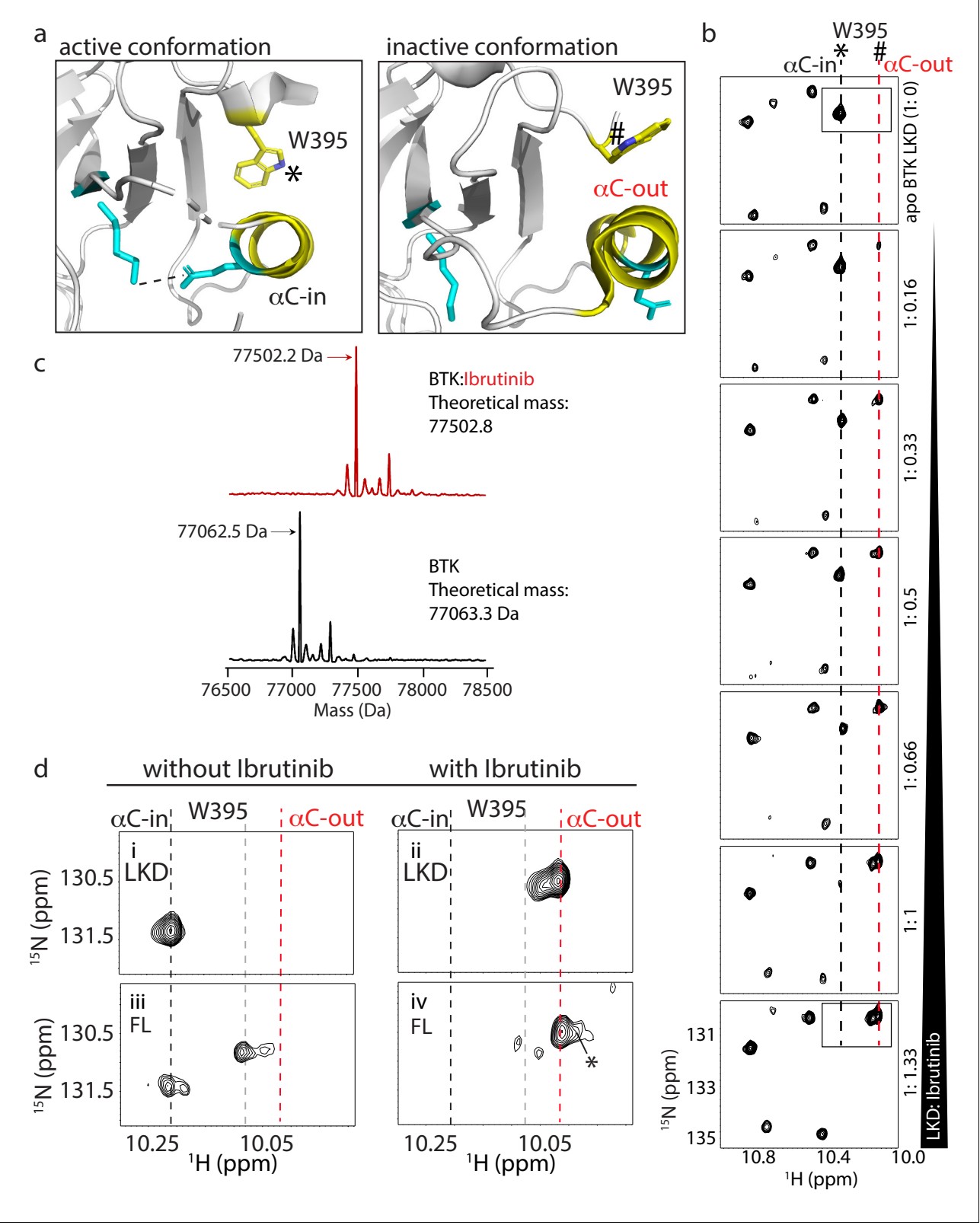

**Figure 2.** W395 indole NH resonance reports on the conformation of BTK. (a) Close-up of the structural differences surrounding the αC helix, W395, and the Lys/Glu salt bridge for structures of the active Bruton's tyrosine kinase (BTK) kinase domain conformation (PDB ID: 3K54) and the inactive BTK kinase domain conformation (PDB ID: 5P9J). W395 and the αC helix are labeled depicted in yellow while the Lys/Glu sides chains are shown in cyan (as in *Figure 1d*). A dashed line between the Lys/Glu pair indicates the presence of the salt bridge between these side chains in the active kinase

*Figure 2 continued on next page*

*Figure 2 continued*

conformation. The downfield resonance in (**b**) corresponds to the W395 indole NH indicated with * and the upfield resonance in (**b**) corresponds to the W395 indole NH indicated with #. (**b**) Titration of Ibrutinib into $^{15}$N-labeled BTK LKD. Tryptophan side chain region of the $^{1}$H-$^{15}$N TROSY HSQC spectrum of BTK LKD shows that addition of increasing concentrations of Ibrutinib (top to bottom) decreases the intensity of the downfield BTK W395 resonance (corresponding to the kinase active (αC-in) conformation, dashed black line), and increases the intensity of the W395 resonance in the upfield position, corresponding to the kinase inactive (αC-out) state (dashed red line). The molar ratio of linker kinase domain (LKD) to Ibrutinib is indicated on the left of each panel. All samples contain the same DMSO concentration. The * and # symbols are defined in (**a**). (**c**) Intact mass analysis of wild-type FL BTK before (bottom spectrum, black) and after 15-min incubation with a twofold molar excess of Ibrutinib (top spectrum, red) showing a mass increase of one Ibrutinib molecule. The peaks corresponding to the mass of BTK or BTK:Ibrutinib are identified with arrows. (**d**) Expanded tryptophan side chain region of the $^{1}$H-$^{15}$N TROSY HSQC spectra showing the resonance(s) of W395 for BTK LKD and FL without (left, *i* and *iii*) and with Ibrutinib (right panels, *ii* and *iv*) The black dashed line in the most downfield position and the upfield red dashed line corresponds to that in (**a**). The gray dashed line indicates the position of the W395 $^{1}$H frequency in FL BTK in the apo inactive (αC-out) state. Asterisk indicate the additional peak (W251) that is evident upon Ibrutinib binding to FL. All NMR samples contain 2% DMSO to ensure the solubility of Ibrutinib. At this concentration, DMSO does not perturb the structure of the protein as comparison of $^{1}$H-$^{15}$N TROSY HSQC spectra of the BTK proteins in the presence or absence of 2% DMSO shows no significant changes (data not shown).

The online version of this article includes the following figure supplement(s) for figure 2:

**Figure supplement 1.** Assignment of W395 indole NH resonance in the inhibitor-bound spectra of BTK linker kinase domain (BTK LKD).

**Figure supplement 2.** NMR data indicate that Ibrutinib binding to BTK FL releases the SH3 domain from the autoinhibited conformation.

drug binding to the kinase active site shifts the conformational preference of the regulatory SH3 domain away from its autoinhibitory contacts on the distal side of the kinase domain. This observation prompted further exploration of the consequences of other BTK active site inhibitors on full-length BTK to determine whether this is a shared feature of active site inhibition of BTK.

## BTK active site inhibitors have different effects on the kinase domain conformation

Crystal structures of the BTK kinase domain have been solved in complex with different active site inhibitors (*Bender et al., 2017*; *Crawford et al., 2018*; *Marcotte et al., 2010*; *Di Paolo et al., 2011*). A comparison of the available structures shows that the majority of inhibitors (e.g. Ibrutinib [irreversible, covalent], GDC-0853 [reversible], CGI1746 [reversible], and CC-292 [irreversible, covalent]) stabilize the inactive or αC-out conformation (*Figure 3a*). In addition to the αC-out conformation, a key structural feature of the inactive kinase domain includes the activation loop collapsed toward the active site burying the Y551 phosphorylation site. In most of the BTK:drug complex structures, the activation loop is resolved but electron density is missing for part of the activation loop in the CC-292-BTK structure (*Figure 3a*). Superposition of these four complexes shows that the overall conformation of the kinase domain is similar regardless of bound inhibitor (*Figure 3b*).

The Dasatinib-bound structure is different; the kinase domain in this complex adopts the active conformation with the αC-helix moved in toward the active site (αC-in) and the activation loop is not visible in the electron density (*Figure 3c*). Superposition of the Dasatinib- and Ibrutinib-bound BTK reveals the conformational change in the αC-helix, which is accompanied by the rotamer shift in the side chain of W395 (*Figure 3d*).

We next compared the tryptophan indole region of the $^{1}$H-$^{15}$N TROSY HSQC spectra of apo BTK LKD to BTK LKD bound to the five different active site inhibitors (*Figure 4a*) and find that binding of GDC-0853 and CGI1746 shifts the conformational equilibrium to the inactive (αC-out) conformation in a manner similar to Ibrutinib (*Figure 4a*). In the inhibitor-bound spectra for GDC-0853 and CGI1746, the W395 indole NH resonance is entirely in the upfield position (*Figure 4a*) and, like the ibrutinib-bound protein, there is no visible peak in the downfield position. These data recapitulate the crystallography; BTK LKD bound to Ibrutinib, GDC-0853, and CGI1746 all stabilize the inactive (αC-out) kinase domain conformation (*Figures 3a* and *4a*). Interestingly, in the GDC-0853-bound BTK LKD spectrum, there is a weak W395 peak adjacent to the major W395 peak (*Figure 4*), indicating that the BTK LKD additionally adopts a minor drug-bound conformation.

The crystal structure of BTK linker-kinase bound to CC-292 reveals clear features associated with the inactive kinase (*Bender et al., 2017*); the αC-helix is positioned out and away from the active site, and the W395 side chain adopts the rotamer conformation consistent with inactive state (*Figure 3a*). However, unlike the Ibrutinib, GDC-0853, and CGI1746 complexes, the activation loop

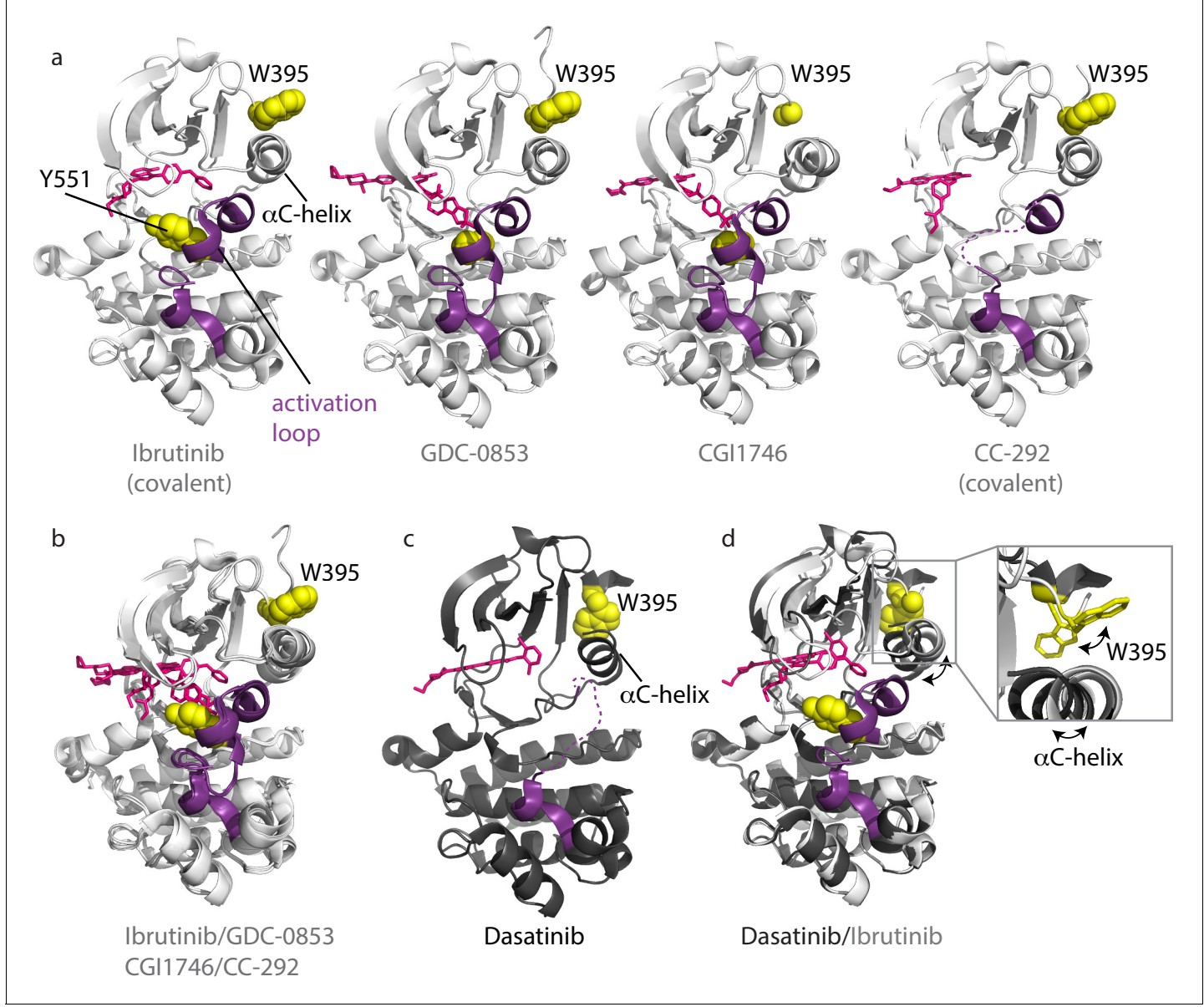

**Figure 3.** Inhibitors stabilize different conformations of the BTK kinase domain. (**a**) Co-crystal structures of BTK linker-kinase domain (light gray cartoon) bound to Ibrutinib (PDB ID: 5P9J), GDC-0853 (PDB ID: 5VFI), CGI1746 (PDB ID: 3OCS), and CC-292 (PDB ID: 5P9L) in the inactive kinase conformation. The inhibitors are shown as pink sticks, the kinase activation loop is purple and the Y551 and W395 residues are shown as yellow spheres. The αC-helix is labeled in the first structure and the two irreversible, covalent inhibitors, Ibrutinib and CC-292, are indicated. Electron density for W395 in the BTK: CGI1746 co-crystal structure and part of the activation loop (purple dashed line) in the BTK:CC-292 co-crystal structure is missing. (**b**) Superposition of the BTK:Ibrutinib, GDC-0853, CGI1746, and CC-292 co-crystal structures shows no major structural variations in the kinase domains. (**c**) Structure of BTK linker-kinase domain (dark gray cartoon) bound to Dasatinib (PDB ID: 3K54) in the active kinase conformation. The activation loop is missing in the structure and is shown as a purple dashed line. Dasatinib is shown as pink sticks, W395 (yellow spheres) and the αC-helix are labeled. (**d**) Superposition of the Dasatinib- (dark gray) and Ibrutinib-bound (light gray) BTK linker-kinase co-crystal structures show the inward movement of the αC-helix and the change in W395 rotamer conformation that accompanies kinase activation (see expanded inset). The BTK:Dasatinib crystal complex was obtained by co-crystallization, whereas the other BTK drug complexes were obtained by soaking BTK crystals with the drug (*Bender et al., 2017*; *Marcotte et al., 2010*).

in the CC-292 crystal structure is not resolved, suggesting that this segment of the protein remains dynamic when bound to CC-292. Comparison of the tryptophan resonances for apo BTK LKD to BTK LKD bound to CC-292 shows the W395 indole NH resonance remains in the downfield position indicating that binding of CC-292 to the active site does not shift the conformational preference of the

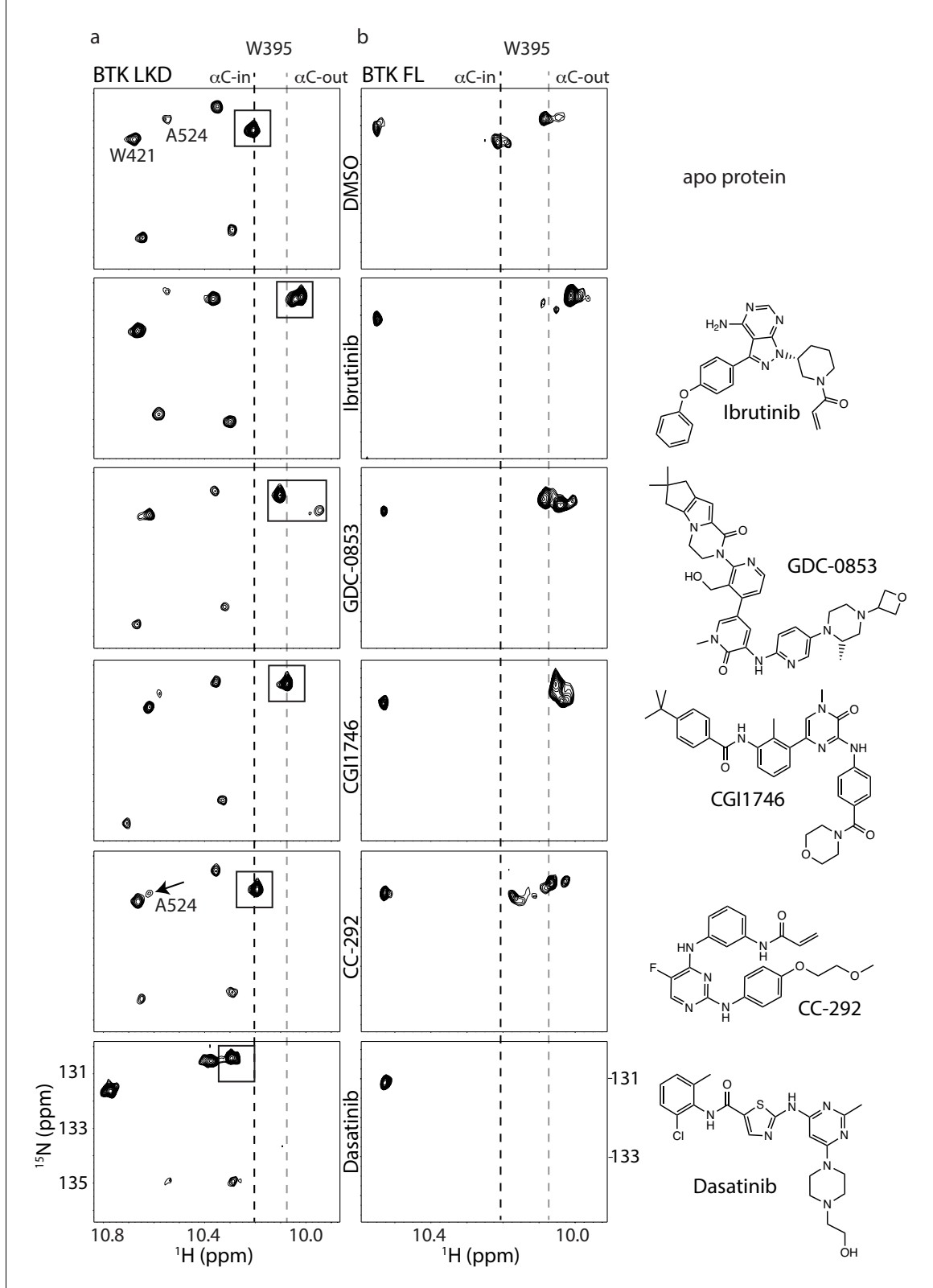

**Figure 4.** Assessing the BTK kinase domain conformational state as a function of inhibitor binding by solution NMR. The structures of each inhibitor are shown on the right. The broken lines show the positions of the BTK W395 resonance in the active (αC-in) and inactive (αC-out) states as in *Figure 2b*. (a) The tryptophan side chain region of the ¹H-¹⁵N TROSY HSQC spectra of ¹⁵N-labeled apo BTK linker-kinase domain (top panel) or bound to various inhibitors (below). The BTK W395 NH resonance (boxed peak) is in the inactive position in the Ibrutinib-, GDC0853-, and CGI1746-bound BTK LKD

*Figure 4 continued on next page*

*Figure 4 continued*

samples. The BTK W395 resonance is maintained in the active, downfield position in the BTK:CC-292 complex and shifted further downfield in the spectrum of the BTK:Dasatinib complex. The chemical shift change observed for A524 resonance in the BTK LKD:CC-292-bound spectrum confirms the binding of CC-292 to BTK LKD despite lack of change in the W395 resonance. (**b**) The tryptophan side chain region of the $^1$H-$^{15}$N TROSY HSQC spectra of $^{15}$N-labeled apo BTK full-length (top panel) or bound to various inhibitors (below). The BTK W395 NH resonance is in the inactive (αC- out) position in the Ibrutinib-, GDC0853-, and CGI1746-bound BTK FL samples. Multiple peaks are observed spanning the active and inactive positions of W395 in the CC-292-bound BTK FL spectrum suggesting that the kinase domain is dynamic and adopts a range of conformations. The W395 resonance is broadened beyond detection in the Dasatinib-bound BTK FL spectrum.

kinase domain in solution toward the αC-out state (*Figure 4a*). Chemical shift perturbations evident elsewhere in the spectrum confirm binding of CC-292 to BTK LKD (*Figure 4a*). Our results suggest that the features of the kinase domain that are trapped by crystallography (*Figure 3a*) do not accurately represent the conformational ensemble of the CC-292-bound kinase domain in solution.

Finally, the crystal structure of Dasatinib-bound BTK (*Marcotte et al., 2010*) is distinct from the other inhibitor-bound structures (*Figure 3b–d*) and the behavior of the W395 resonance is consistent with the crystal structure. Crystallography and NMR both indicate Dasatinib-bound BTK adopts the active (αC-in) conformation (*Figure 4a*). Dasatinib binding causes a further downfield shift in the $^1$H resonance frequency of W395 suggesting that either binding of this drug increases the population of the αC-in conformation compared to apo BTK LKD or may be reflective of local chemical shift changes due to drug binding to the active site. Acquisition of separate $^1$H-$^{15}$N TROSY HSQC spectra for Ibrutinib, GDC-0853, CGI1746, CC-292, and Dasatinib bound to the BTK linker-kinase (W395A) mutant confirms the W395 assignments (*Figure 2—figure supplement 1*).

To test whether the conformational states of the BTK LKD complexed to the different drugs are similar for full-length BTK, we analyzed the corresponding NMR spectra of full-length BTK bound to each of the drug molecules. Binding of Ibrutinib, GDC-0853, and CGI1746 all shift the conformational equilibrium of the kinase domain in the full-length protein to the αC-out conformation in a manner that is nearly identical to drug binding to the shorter LKD fragment (*Figure 4b*). Although the magnitude and direction of the chemical shift changes are the same for the fragment and full-length BTK, the inhibitor-bound W395 indole NH resonances in these full-length BTK complexes are more complex with multiple overlapping peaks likely reflecting conformational heterogeneity around the αC-out state of the multi-domain, full-length protein (*Figure 4a and b*).

Despite its covalent binding to the active site, CC-292 bound to full-length BTK highlights the dynamic nature of the bound complex that is partially evident in the BTK:CC-292 crystal structure (*Figure 3a*). The $^1$H-$^{15}$N TROSY HSQC spectrum of full-length BTK bound to CC-292 gives rise to a range of W395 indole NH resonances corresponding to both the αC-in and αC-out states (*Figure 4b*). It is likely that the crystal structure of BTK LKD bound to CC-292 captured the lowest energy, inactive conformation, whereas solution NMR methods reveal the extent of conformational sampling of the CC-292/BTK complex.

And last, Dasatinib binding to full-length BTK leads to extensive line-broadening throughout the spectrum. The W395 resonance is no longer detected but other resonances remain visible (*Figure 4b*). The observed line-broadening likely reflects increased dynamics of the conformational ensemble upon Dasatinib binding to the active site. While distinctly different from one another, the BTK:Dasatinib and BTK:CC-292 complexes both exhibit increased dynamic motions in the kinase domain compared to BTK bound to Ibrutinib, GDC-0853, or CGI1746. To complement the NMR data (which reveals the conformational status of just the kinase domain), we next sought further details about the effects of drug binding on the regulatory domains within the full-length protein by subjecting the BTK:drug complexes to hydrogen-deuterium exchange mass spectrometry (HDX-MS).

## Dramatic differences in full-length BTK:drug complexes revealed by HDX-MS

We have previously established that HDX-MS can be used to assess the conformational status of each domain in full-length BTK as the conformational equilibrium shifts between the 'closed' autoinhibited and the 'open' active states (*Joseph et al., 2017*). In solution, full-length BTK exists predominantly in the 'closed' autoinhibited conformation (*Figure 1b*). BTK activating mutations that destabilize the autoinhibitory conformation result in an increase in deuterium uptake, compared to

wild-type BTK, in peptides derived from the SH3, SH2, and SH2-kinase linker region (*Joseph et al., 2017*). Building on this approach, full-length BTK was mixed separately with either a twofold molar excess of Ibrutinib or a 10-fold molar excess of GDC-0853, CGI1746, CC-292, or Dasatinib and each sample was subjected to HDX-MS analysis. We compared the effects of each inhibitor on full-length BTK using a set of peptic peptides common among all six forms (apo and five inhibitor-bound states, the identities of this subset of peptides are listed in *Source data 1*) which report on the conformational status of the regulatory domains and the kinase domain.

Inhibitor binding to full-length BTK causes clear effects on conformation and dynamics as measured by HDX-MS, and there are stark differences between the different inhibitors (*Figures 5* and *6*). Comparison of deuterium uptake for Ibrutinib conjugated to full-length BTK with that of apo BTK (*Figures 5a* and *6*) shows that Ibrutinib labeling of C481 of BTK FL alters deuterium exchange in several different regions of the protein. As expected, the β2 and β3 strands in the kinase N-lobe as well as the β7, β8 strands and the N-terminus of the kinase activation segment (DFG motif) show reduced deuterium incorporation that is consistent with the location of Ibrutinib binding in the kinase active site (*Figure 6*). Importantly, despite Ibrutinib stabilizing an inactive (αC-out) conformation in the kinase domain, *increases* in deuterium exchange are observed in both SH3 and SH2 domains as well as in the SH2-kinase linker, consistent with destabilization of the autoinhibitory interface (*Figures 5a* and *6*). Similar increases in deuterium uptake in BTK regulatory elements have been observed for mutations that shift the equilibrium away from the autoinhibited BTK conformation and increase activity (*Joseph et al., 2017*). Together, the NMR and HDX-MS data indicate that Ibrutinib binding to the active site favors a 'hybrid' conformational state of full-length BTK: an *inactive* (αC-out) conformation is stabilized in the kinase domain, while the regulatory domains are at least partially released from the distal surface of the kinase domain to populate an open '*active*' conformation.

The allosteric effects of Ibrutinib binding to the BTK active site are quite distinct from the effects induced by GDC-0853 and CGI1746 (*Figures 5a,b,c* and *6*). Despite the strong similarities between the W395 resonance frequencies (*Figure 4*) and the crystal structures of Ibrutinib, GDC-0853, and CGI1746 bound to the BTK kinase domain (*Figure 3a,b*), the HDX-MS data show that GDC-0853 and CGI1746 lead to pronounced protection from exchange in the kinase domain but they do not affect the regulatory domains (*Figures 5b,c* and *6*). Protection induced by GDC-0853 and CGI1746 in the regions surrounding the active site is similar to that observed for Ibrutinib albeit greater in magnitude. Specifically, upon binding of GDC-0853 and CGI1746 the activation loop (e.g. *Figure 5b,c*, peptide iv) is more strongly protected than when bound to Ibrutinib while parts of the αF and αG helices are strongly protected with GDC-0853 and CGI1746 but not protected at all by Ibrutinib (*Figures 5a,b,c* and *6*). It is also important to note that peptides that cover β1–3 in the N-lobe of the kinase domain (e.g. *Figure 5*, peptide iii) are both sensitive to the presence of the inhibitor (protection from exchange in the inhibitor-bound forms compared to the apo-form) and see an attenuation in the degree of that protection (difference between BTK:Ibrutinib and BTK: GDC-0853/CGI1746) as the regulatory domains are released from the autoinhibited conformation. While the HDX-MS changes within the kinase domain are consistent with the binding mode and contacts being made by the drug within the BTK active site in the BTK:drug co-crystals (*Figure 3b*), HDX-MS of full-length BTK reveals that there are clear conformational differences in the BTK regulatory domains between the drug-bound forms in solution. An extension of this type of drug-dependent allostery has been recently reported in the context of a dual-liganded enzyme (*Ghode et al., 2020*).

The kinase domain conformation of the BTK:CC-292 complex shares structural similarity with the Ibrutinib, GDC-0853, and CGI1746 BTK complex crystal structures but the solution behavior of the CC-292 complex points to differences. While CC-292 is covalently attached to BTK C481 and contacts the hinge region of the kinase, it does not extend further into the kinase active site setting it apart from the other BTK inhibitors (*Figure 3*). Consistent with this, HDX-MS reveals only limited protection upon CC-292 binding localized primarily to the N-lobe surrounding the active site (*Figure 5d*, peptide iii, 6). The lack of protection from deuterium exchange in the activation loop of BTK upon binding to CC-292 is consistent with high flexibility and the absence of electron density for this region in the crystal structure of CC-292 bound to BTK LKD (*Figure 3a*). The smaller and more confined HDX differences between apo- and CC-292-bound full-length BTK are consistent with the similar conformational modulation evident in the W395 resonance for apo BTK and BTK:CC-292 (*Figure 4b*). Finally, the BTK:CC-292 complex, like the GDC-0853 and CGI1746 complexes, is

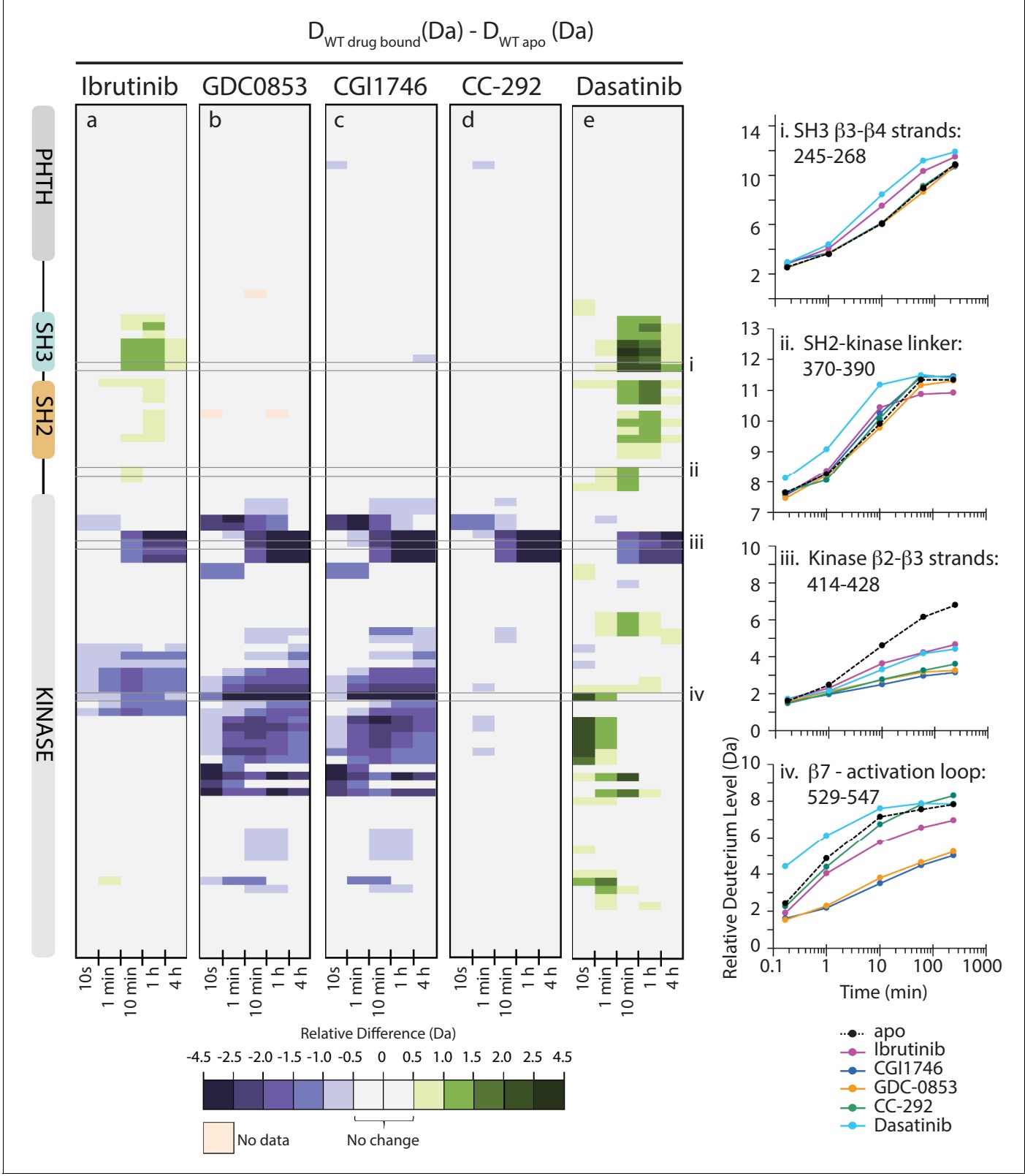

**Figure 5.** Active site inhibitors induce different conformational responses in full-length BTK. (a–e) The measured relative deuterium level of peptides in apo BTK was subtracted from the deuterium level of the corresponding peptide from each drug-bound form of BTK ($D_{drug-bound}$-$D_{apo}$) and the differences colored according to the scale shown. In this and subsequent figures, peptic peptides are shown from N- to C-terminus, top to bottom, and the amount of time in deuterium is shown left to right. The relative difference data shown here represents a curated set of peptides that are coincident

*Figure 5 continued on next page*

*Figure 5 continued*

across all six states (apo and five drug-bound BTK forms). The identification of these chosen peptides, the relative difference values, and the complete data set for each state can be found in *Source data 1*. The approximate position of the domains of BTK, as described in *Figure 1a*, is shown at the left. Deuterium uptake curves of selected peptides (indicated with a gray box in panels a-e and labeled i-iv) from various regions of the protein are shown on the right.

characterized by a lack of changes in the regulatory domains. As full-length BTK is predominantly in the autoinhibited conformation in solution (*Joseph et al., 2017*), this result indicates that binding of GDC-0853, CGI1746, and CC-292 do not shift the equilibrium away from the inactive, autoinhibited conformation of full-length BTK.

HDX-MS of the BTK:Dasatinib complex shows that this complex exhibits the largest differences compared to apo full-length BTK (*Figures 5e* and *6*). In contrast to the other BTK:drug complexes, Dasatinib binding causes dramatic increased deuterium uptake in a large part of the kinase domain C-lobe and activation loop (*Figure 5e*, peptide iv, 6), a result suggesting that these regions of the protein become less structured and more solvent exposed upon binding to Dasatinib. Such changes are consistent with the extensive NMR line broadening (*Figure 4b*) and the absence of electron density for the activation loop in the crystal structure of Dasatinib bound to BTK (*Figure 3c*). In addition, the SH3 and SH2 domains and the SH2-kinase linker region all exhibit increased deuteration upon Dasatinib binding to the BTK active site (*Figures 5e* and *6*) and the extent of deuterium uptake is greater than that observed for Ibrutinib binding suggesting Dasatinib shifts the population away from the autoinhibited form to a greater extent than Ibrutinib (*Figure 5a,e*, peptide i and 6). Indeed, the W395 resonance suggests that the kinase domain in the Dasatinib:BTK complex populates the αC-in conformation rather than the αC-out conformation stabilized by Ibrutinib providing a possible explanation for how Dasatinib induces greater exposure in the regulatory domains. Taking the results for all the inhibitors together, NMR and HDX-MS data provide unique insights into the extent to which different BTK inhibitors affect the conformational preferences and dynamics of both the kinase domain itself and the regulatory domains within full-length BTK. Ibrutinib and Dasatinib promote changes in the conformational equilibria of full-length BTK, whereas the other inhibitors, GDC-0853, CGI1746, and CC-292, cause significantly more localized conformational adjustments limited to the kinase domain.

## The Ibrutinib resistance mutation T316A activates BTK by disrupting the autoinhibitory conformation of full-length BTK

Refocusing on the clinically important BTK inhibitor, Ibrutinib, we turn next to evaluating the cause of Ibrutinib resistance upon mutation of BTK T316 to alanine. Unlike the majority of Ibrutinib resistance mutations, including the most common C481S mutation in the kinase domain active site (*Furman et al., 2014*; *Woyach et al., 2014*; *Young and Staudt, 2014*; *Cheng et al., 2015*; *Zhang et al., 2015*), BTK T316A is located outside of the kinase domain in the BTK SH2 domain (*Figure 1b*; *Sharma et al., 2016*; *Kadri et al., 2017*). The location of T316 suggests that mutation at this position might destabilize autoinhibitory contacts between the SH2 domain and the kinase domain C-lobe. T316 is adjacent to R307, a key residue in the autoinhibitory salt bridge interaction between the SH2 domain and D656 in the C-terminus of the kinase domain (*Figure 7a, b*; *Joseph et al., 2017*). Based on the crystal structure of this region of autoinhibited BTK (*Wang et al., 2015*), T316 is predicted to stabilize the productive autoinhibitory conformation of the R307 sidechain via a network of hydrogen bonds, hydrophobic, and cation-pi contacts involving R288, R307, S309, H333, D656, and E658 (*Figure 7a,b*). Loss of even the small threonine side chain could be deleterious to this local structure and lead to a conformational change that is propagated across the autoinhibitory interface involving the entire SH3-SH2-linker region, thereby activating BTK.

To more fully understand the consequences of the T316A mutation on BTK and to gain insight into how a mutation that is remote from the active site confers resistance to Ibrutinib, we applied NMR and HDX-MS approaches to the full-length BTK T316A mutant. Comparing the extent of deuterium uptake for wild-type BTK versus the T316A mutant shows small increases in exchange in the non-catalytic regulatory SH3-SH2 region upon mutation of T316 to alanine (*Figure 7c,d*). Mapping the peptides that show increased deuterium uptake for the BTK T316A mutant onto the

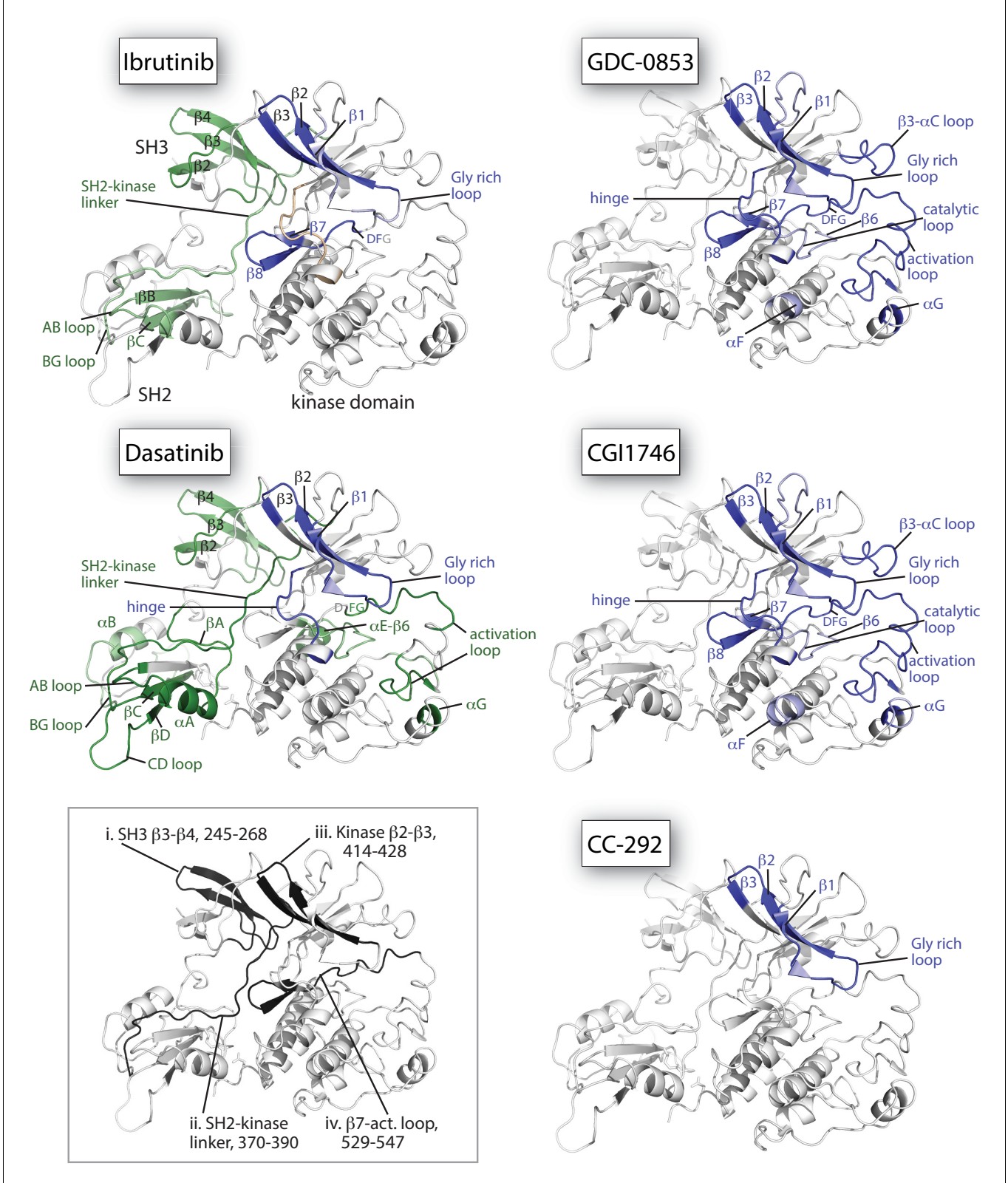

**Figure 6.** Mapping HDX-MS changes induced by each drug on the structure of the BTK SH3-SH2-kinase fragment (PDB ID: 4XI2). For the five drug:BTK complexes the colors from *Figure 5* are used: major differences greater than 1.0 Da are shown as dark blue (decrease) or dark green (increase); modest differences between 0.5 Da and 1.0 Da are shown as light blue (decrease) and light green (increase). Localization of the changes in deuterium

*Figure 6 continued on next page*

*Figure 6 continued*

incorporation was accomplished using overlapping peptides included in the complete peptide data set provided in *Source data 1*. The inset shows the location of peptides i – iv from *Figure 5* (each peptide is black and labeled with residue numbers spanning the N- to C-termini of the peptic fragment).

autoinhibited structure shows that the regions affected by the mutation in the SH2 domain localize to the SH3-SH2 domains and SH2-kinase linker that together create autoinhibitory contacts with the distal side of the kinase domain (*Figure 7e*). It should be noted that these small 0.5–1.0 Da changes cannot be specifically localized within the regions indicated and so the peptides shown encompass a region larger than the actual change in deuteration. As well, the changes observed for the T316A mutation are consistent with a more transiently populated 'open' conformation compared to the more drastic changes in deuterium uptake observed upon mutation of D656 in previous work (*Joseph et al., 2017*). Loss of the D656 sidechain abolishes the autoinhibitory salt bridge with R307 (*Figure 7b*) and leads to activation of BTK (*Joseph et al., 2017*). In contrast to the changes in deuterium exchange observed for the BTK T316A mutant, the BTK C481S active site resistance mutant shows no change in exchange behavior compared to the wild-type protein (*Figure 7d*, *Source data 1*).

NMR analysis confirms that the T316A mutation in the BTK SH2 domain shifts the conformational ensemble of BTK toward the active kinase (αC-in) population. Comparing the $^1$H-$^{15}$N TROSY HSQC spectrum of the full-length BTK T316A to wild-type BTK shows a change in the conformational equilibrium upon mutation (*Figure 8a*). Peak intensities for the W395 resonance in the upfield and downfield positions indicate that 72% of full-length wild-type BTK adopts a conformation consistent with inactive kinase (αC-out) and mutation of T316 to alanine results in only 60% of the full-length kinase in an inactive conformation. Accordingly, the population of active BTK as measured by W395 peak intensity increases ~1.5-fold, from 28% for wild-type BTK to 40% for the BTK T316A mutant (*Figure 8a*). Consistent with the HDX-MS data (*Figure 7c*), the partial shift in population to the αC-in state in the T316A mutant is small when compared to the the BTK D656K activating mutant, which shows an almost complete shift in the conformational ensemble to the active (αC-in) state (*Joseph et al., 2017*). To directly test kinase activity, we compared autophosphorylation of BTK wild-type and T316A mutant under identical kinase assay conditions. Phosphorylation on Y551 in the BTK T316A activation loop is twice that of wild-type BTK consistent with the shift in the conformational ensemble measured by NMR (*Figure 8b,c*). BTK C481S mutant is included the kinase assay and, as predicted by the identical deuterium exchange behavior of this mutant compared to wild-type, no difference in kinase activity is observed between wild-type BTK and the C481S mutant (*Figure 8b,c*). Thus, the remote T316A resistance mutation in the BTK SH2 domain leads to an increase in the population of active enzyme that is reflected by an increase in activity.

## Ibrutinib binds to the BTK T316A resistance mutant

The shift in the conformational ensemble of BTK T316A away from the autoinhibited conformation could give rise to a conformational state that disfavors Ibrutinib binding, thereby contributing to Ibrutinib resistance. We examined Ibrutinib conjugation to the BTK T316A mutant by following the W395 resonance and the HDX-MS signatures. Both experimental methods require protein concentrations in the micromolar range and so it is noted that the experiments are carried out well above the nanomolar IC$_{50}$ values previously reported for Ibrutinib (*Furman et al., 2014*; *Woyach et al., 2014*; *Hamasy et al., 2017*). W395 reports a shift toward the inactive (αC-out) kinase domain conformation upon Ibrutinib binding to the BTK T316A mutant that mirrors the results of Ibrutinib binding to wild-type BTK (*Figure 8d*). Ibrutinib binding to the T316A mutant results in increased deuterium exchange in the region spanning the SH3, SH2 domains and SH2-kinase linker in a manner that is nearly identical to wild-type protein (*Figure 8e*, *Figure 8—figure supplement 1*). Thus, the Ibrutinib-induced remote conformational changes in the regulatory region of BTK appear to be a general consequence of Ibrutinib binding that occurs regardless of the T316A mutation in the BTK SH2 domain.

The covalent nature of Ibrutinib binding to the BTK active site prevents standard measurement of affinity constants. To assess whether the T316A mutation in the SH2 domain affects drug binding to the active site, we instead performed an HDX-MS titration series to compare Ibrutinib binding to

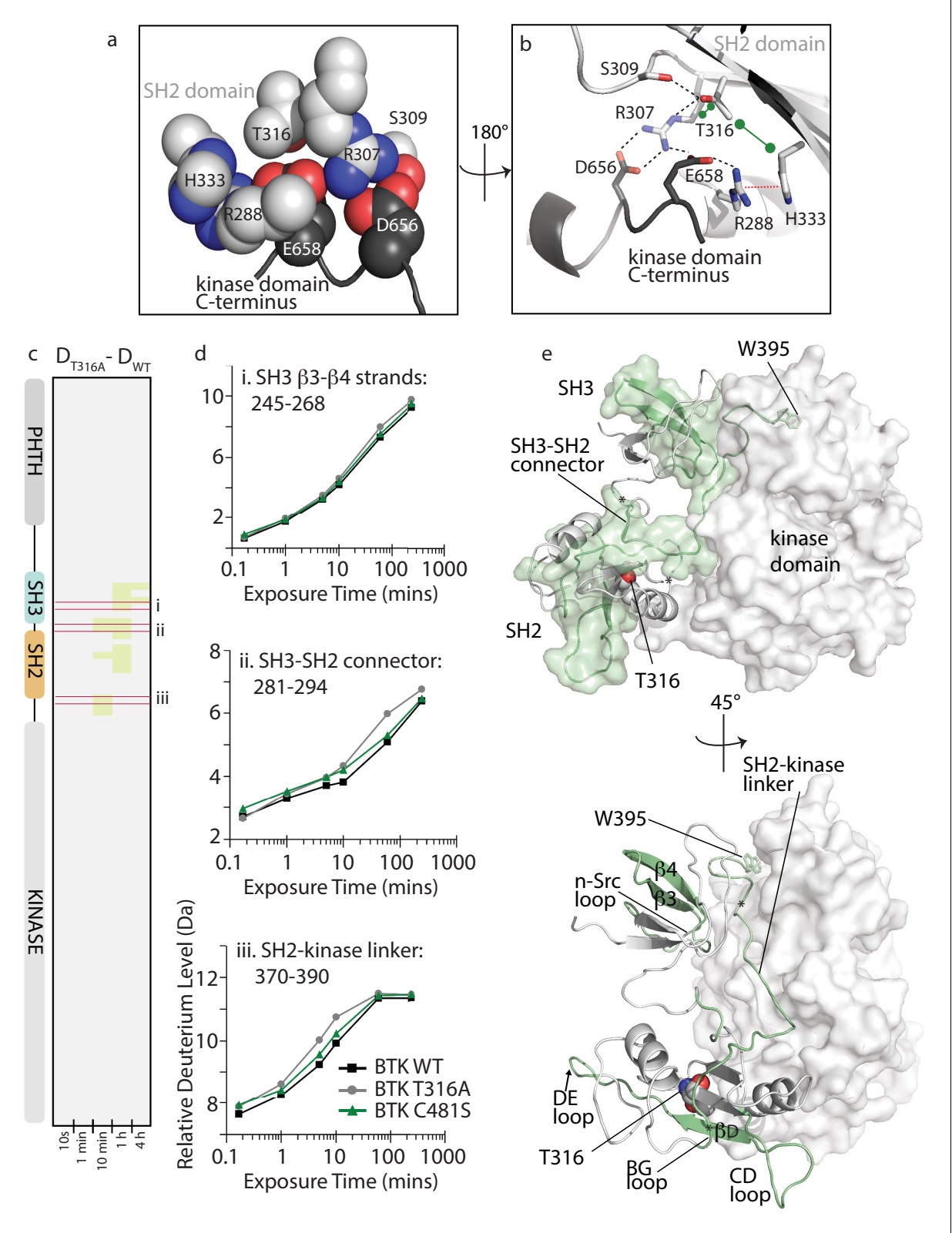

**Figure 7.** BTK T316A mutation disrupts the autoinhibitory conformation of BTK. (**a,b**) Close-up views of the BTK 32LKD structure (PDB ID: 4XI2) illustrating the network of interactions involving the side chain of T316. In (**a**), the sidechains are depicted in spheres and labeled, all SH2 sidechains are in light gray, and the two sidechains from the kinase domain C-terminus are in dark gray. Using the same color scheme, the structure is rotated in (**b**) and shows the sidechains in stick format as well as the cartoon of this part of the SH2 domain structure. Black dashed lines indicate potential for

*Figure 7 continued on next page*

*Figure 7 continued*

hydrogen bonds, the red dashed line indicates a cation-pi interaction, and the green dumbbell indicates potential for hydrophobic packing. (**c**) HDX difference data for the BTK T316A mutant ($D_{T316A}$-$D_{WT}$). Color scale and peptide/time course arrangement are the same as in *Figure 5*. See *Source data 1* for additional information, including all peptide identifications and deuterium values. (**d**) Deuterium uptake curves from peptides indicated by red boxes in panel c, (labeled i, ii, and iii), which show an increase in deuterium uptake in the BTK T316A mutant versus BTK WT. (**e**) Two views of the structure of BTK 32LKD (PDB ID: 4XI2) showing the regions of increased deuteration (green surface and cartoon) upon T316A mutation. T316 sidechain in the BTK SH2 domain is indicated in spheres. Specific regions of secondary structure in the SH3 and SH2 domains that are more deuterated upon T316A mutation are labeled in the lower panel.

wild-type and T316A BTK. No meaningful differences (>0.5 Da) in deuterium uptake are observed between wild-type BTK and the BTK T316A mutant over a range of Ibrutinib concentrations (*Figure 8—figure supplement 2*). Thus, our results indicate that the BTK T316A mutation likely causes resistance to Ibrutinib (*Sharma et al., 2016*) by shifting the conformational ensemble of full-length BTK to favor the active state. At concentrations of Ibrutinib that exceed the $IC_{50}$, there is no evidence that the T316A mutation in the SH2 domain limits binding of drug to the BTK active site.

## Discussion

The essential role that kinases play in cell signaling make them attractive pharmacological targets (*Zhao and Bourne, 2018*; *Müller et al., 2015*). The vast majority of available kinase inhibitors target the kinase active site and our work shows that the occupancy of the kinase active site by different inhibitors can have profoundly distinct effects on the distant BTK regulatory domains, effects that are missed in crystal structures of drug-bound kinase domain alone. Here, we provide the first report of the impact of binding of the irreversible FDA-approved drug Ibrutinib on the conformation of full-length BTK. Approval of an increasing number of covalent inhibitors demonstrate the effectiveness of irreversible drugs compared to reversible inhibitors in the targeted therapies of kinases (*Zhao and Bourne, 2018*; *Bauer, 2015*). A major advantage of irreversible inhibitors is that once covalently bound to the kinase, they permanently trap the protein, thereby sequestering its catalytic function until fresh protein is produced by translation (*Barf and Kaptein, 2012*). Consequently, covalent inhibitors are particularly effective in the treatment of kinases that have a long cellular half-life (*Bauer, 2015*). Additionally, covalent inhibitors are selected for rapid clearance from the serum to prevent toxicity effects arising from off-target modification (*Barf and Kaptein, 2012*). Ibrutinib levels in the serum peak 1–2 hr post-ingestion and return to basal levels after 3–4 hr (*Advani et al., 2013*). The BTK C481S resistance mutation prevents the covalent attachment of Ibrutinib to the active site during the period when Ibrutinib is available in the serum but Ibrutinib still binds noncovalently (*Furman et al., 2014*). This reversible drug binding coupled with rapid serum clearance implies that the BTK C481S mutant is not 'trapped' by Ibrutinib. Once serum Ibrutinib levels diminish, BTK signaling by the BTK C481S mutant occurs resulting in drug resistance.

The BTK T316A resistance mutant retains the ability to be covalently modified by Ibrutinib. Thus, when the BTK T316A mutant encounters Ibrutinib it is likely 'trapped' by the covalent drug in a manner similar to wild-type BTK. However, the BTK T316A mutation is itself activating; loss of the T316 sidechain destabilizes a key down-regulatory interaction within BTK and shifts the conformational equilibrium away from the autoinhibited state (*Figures 7c* and *8a*). And so, newly synthesized BTK T316A could escape the narrow inhibition time window created by rapid Ibrutinib clearance. The unconjugated and active T316A mutant could then promote BTK signaling, unlike properly autoinhibited wild-type BTK. The idea that resistance mutations can counteract the autoinhibitory state has been elegantly demonstrated in recently published work on the ABL kinase (*Xie et al., 2020*). It should be noted that we have not characterized binding of Ibrutinib to the phosphorylated form of BTK and so it is possible that phosphorylated BTK T316A could exhibit diminished affinity for Ibrutinib thereby also contributing to the observed resistance.

Our findings now show that mutation is not the only mechanism by which the BTK conformational ensemble shifts away from the 'closed' autoinhibited state. Binding of either Ibrutinib or Dasatinib triggers the same shift in the conformational equilibrium of the regulatory domains away from the autoinhibited conformation creating the opportunity for BTK regulatory domain interactions with exogenous ligands despite the inhibited kinase domain. This conformational shift could promote

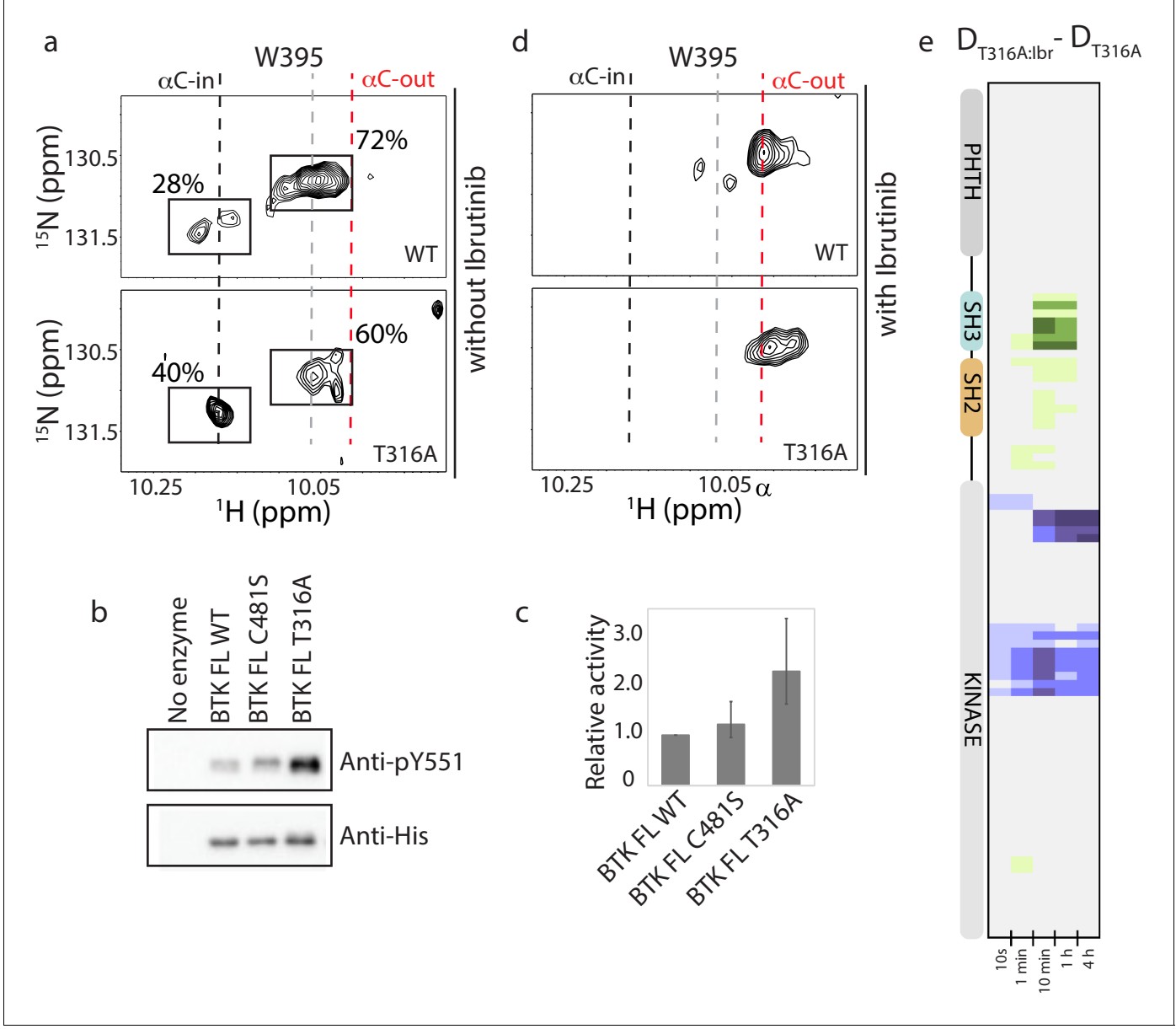

**Figure 8.** BTK T316A mutation increases the population of active BTK. (a) Tryptophan side chain region of the $^1$H-$^{15}$N TROSY HSQC spectra of full-length BTK WT and T316A in the absence of Ibrutinib and DMSO. The black, gray and red dashed lines are as described for *Figure 2d*. The relative intensities of the peaks corresponding to the active (αC-in) and inactive (αC-out) kinase domain conformations are indicated as percentages. (b) Western blot comparing the kinase activity of full-length (FL) BTK wild-type, C481S and T316A mutants. BTK autophosphorylation monitored using the BTK pY551 antibody and the total protein levels monitored using the Anti-His antibody. (c) Histogram quantifying the western blots shown in (b). The blots were quantified and normalized as described in the Materials and methods. Data shown are the average of three independent experiments. Phosphorylation levels of FL BTK WT, C481S, and T316A on Y551 were undetectable by western immuno-detection prior to the start of the activity assay. (d) Tryptophan side chain region of the $^1$H-$^{15}$N TROSY HSQC spectra of full-length BTK WT and T316A in the presence of Ibrutinib. (e) HDX difference data upon Ibrutinib binding to BTK T316A ($D_{T316A:Ibr}-D_{T316A}$). Color scale and peptide/time course arrangement are the same as in *Figure 5*. See *Source data 1* for additional information, including all peptide identifications and deuterium values. A side-by-side comparison of data in (e) with data in *Figure 5* is provided in *Figure 8—figure supplement 1*.

The online version of this article includes the following figure supplement(s) for figure 8:

**Figure supplement 1.** Side-by-side comparison of HDX-MS data from *Figure 5* and *Figure 8*.

**Figure supplement 2.** HDX-MS Ibrutinib titration into FL BTK wild-type or T316A.

kinase-independent signaling events (*Saito et al., 2003*; *Middendorp et al., 2003*; *Middendorp et al., 2005*) and/or create a dominant negative. Varied effects of inhibitor binding on the non-catalytic functions of kinases have been observed previously and range from alterations in interactions with upstream or downstream regulatory kinases and phosphatases to changes in ligand affinity (*Sonti et al., 2018*; *Leonard et al., 2014*; *Tong et al., 2017*; *Skora et al., 2013*). For BTK specifically, recent evidence suggests that non-catalytic functions of Ibrutinib-bound BTK activate CLL-specific PLCγ variants (*Wist et al., 2020*). Thus, selection of BTK inhibitor type to treat specific disease states as well as design of the next generation of BTK inhibitors need to carefully consider the impact of the inhibitor on BTK regulatory domain conformation.

Of the five active site inhibitors studied here only CGI1746, CC-292, and GDC-0853 inhibit kinase activity without disrupting the autoinhibitory conformation of the full-length kinase. Ibrutinib and Dasatinib alter the conformational equilibrium of the multidomain kinase decreasing the population of the autoinhibited kinase even though the drugs block and inactivate the kinase domain. The effect of an active site inhibitor on the overall conformational preference of a multidomain kinase must originate in the kinase domain. Multiple mobile elements including the αC-helix, DFG motif, and the activation loop switch the kinase domain between its inactive and active conformations (*Taylor et al., 2015*; *Taylor and Kornev, 2011*; *Fabbro et al., 2015*). A number of studies are emerging that focus on the long-range conformational effects of active site inhibitor binding to the SRC module kinase families: SRC, ABL, and TEC. There is an overriding effort to define a unifying metric that can predict the outcome of inhibitor binding on the conformation of the regulatory domains and different studies have endeavored to link disassembly of the regulatory domains to different mobile elements within the kinase domain.

For the SRC family, the αC-in configuration has been correlated with release of the regulatory domains from the autoinhibited state (*Leonard et al., 2014*; *Tsutsui et al., 2016*). Other studies suggest that the αC-helix conformation alone can be used to predict the regulatory domain conformation in the SRC, ABL, and TEC kinases; inhibitors that stabilize the αC-in conformation disrupt the regulatory domains from their autoinhibitory conformation, whereas αC-out stabilizing inhibitors do not (*Fang, 2020*). However, ABL focused studies show that Dasatinib stabilizes αC-in but does not disrupt the regulatory domains from their compact, autoinhibitory state (*Sonti et al., 2018*). For ABL, a strict correlation has been observed instead between activation loop conformation and regulatory domain assembly (*Sonti et al., 2018*; *Skora et al., 2013*). Inhibitors that stabilize an activation loop 'in' conformation disrupt the ABL SH3 and SH2 domains from their autoinhibited conformation, whereas inhibitors that stabilize the activation loop 'out' conformation do not. Our results suggest simple αC-helix or activation loop 'in' versus 'out' metrics are not sufficient to explain BTK inhibitor effects. Just one example from the data presented here shows that both Ibrutinib and CGI1746 favor the activation loop 'in' configuration, but the regulatory domains are displaced upon Ibrutinib binding and remain in the autoinhibited conformation on binding CGI1746.

To understand why Ibrutinib and Dasatinib disrupt the autoinhibited conformation of full-length BTK while GDC-0853, CGI1746, and CC-292 do not, we turn back to the crystal structures of BTK LKD bound to each inhibitor (*Figure 3*). Probing for features that are common to the Ibrutinib- and Dasatinib-bound structures but distinct from the other inhibitor-bound structures, we observed two distinct drug orientations among the five BTK:drug complexes (*Figure 9a,b*). Interestingly, the orientations of Ibrutinib and Dasatinib (the two inhibitors that disrupt the full-length BTK autoinhibitory structure), differ significantly from that of GDC-0853, CGI1746, and CC-292 (*Figure 9a,b*). Ibrutinib and Dasatinib are oriented toward the 'back pocket' of the kinase active site (also called the back cleft or hydrophobic pocket II (HPII)) (*Roskoski, 2016*), whereas GDC-0853, CGI1764, and CC-292 bypass the back pocket and extend into the front pocket toward the activation loop (*Figure 9a,b*). All five compounds maintain contacts with the kinase hinge region (*Figure 9a*).

In accessing the back pocket of the active site, Ibrutinib and Dasatinib alter the back pocket around the regulatory (R-) spine (*Taylor et al., 2015*; *Kornev and Taylor, 2010*; *Kornev et al., 2008*) residue L460 (*Figure 9c*). Ibrutinib directly contacts L460, while Dasatinib binding induces the inward movement of the αC-helix bringing M449 in direct contact with L460 (*Figure 9c*). In contrast, the compounds that extend into the front pocket, such as GDC-0853 (*Figure 9c*), avoid all contact, direct or indirect with L460. L460 is adjacent to Y461, a component of the conserved 'hydrophobic stack'. The 'hydrophobic stack' consists of two residues on the distal surface of the kinase domain (Y461 and W421 in BTK) and a single hydrophobic residue from the linker spanning the SH2 and

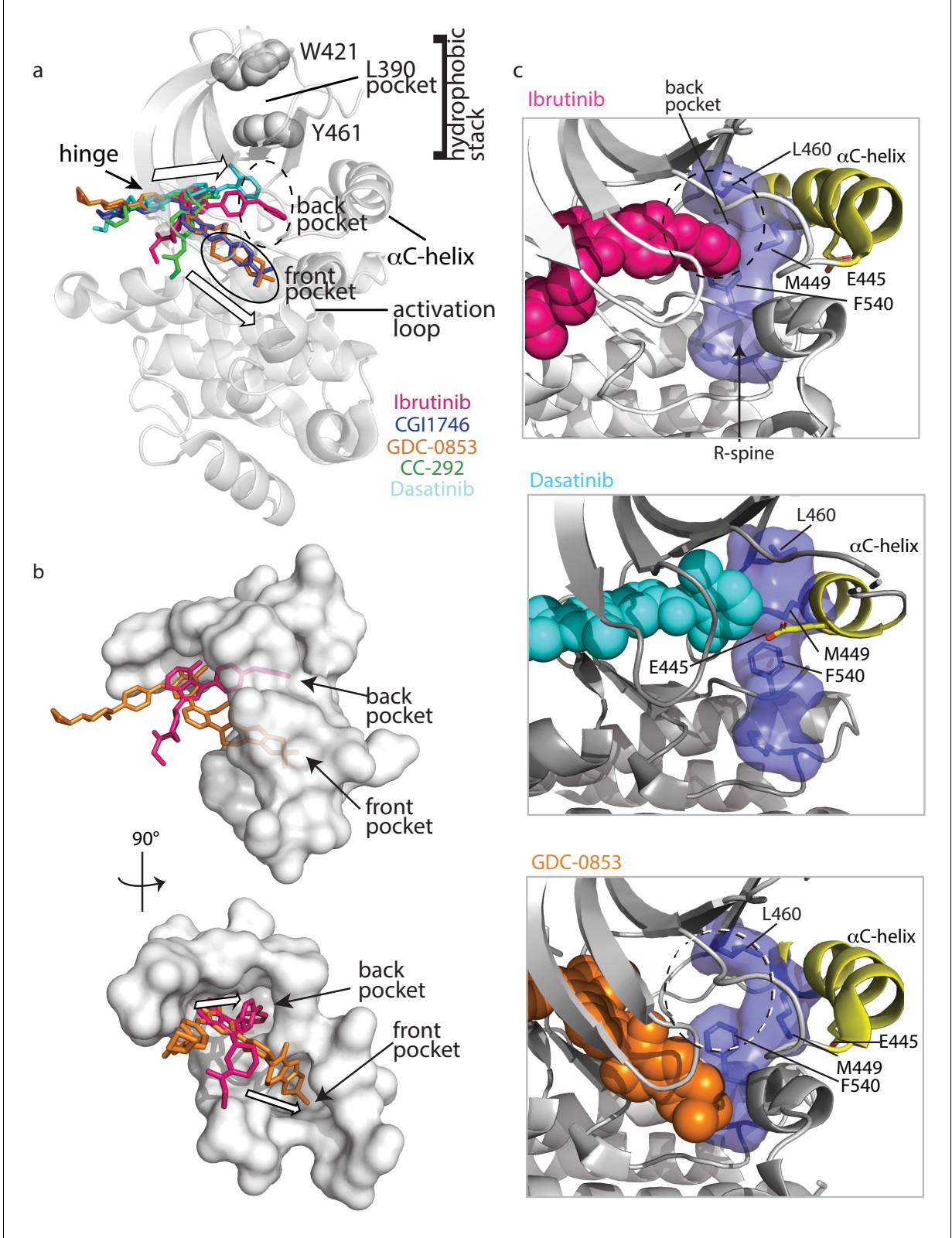

**Figure 9.** Ibrutinib and Dasatinib occupy the back pocket of the BTK active site. (a) Superposition of the five inhibitor-bound structures of BTK linker-kinase domain (gray cartoon) show that Dasatinib and Ibrutinib are oriented toward the back of the kinase and GDC-0853, CGI1746, and CC-292 fill the front pocket. The five drugs are displayed as sticks and colored as in *Figure 5*, and only one kinase domain is included for clarity. (b) An enlargement of the region surrounding the active site, with surface rendering of the active site pockets. Two views are shown and only Ibrutinib and GDC-0853 are

*Figure 9 continued on next page*

Figure 9 continued

included for clarity as they represent the two binding modes. (c) Close-up view of the BTK active site with Ibrutinib (top), Dasatinib (middle), and GDC-0853 (bottom) depicted in spheres. L460 lines the kinase back pocket. The R-spine is shown in blue sticks and transparent surface with L460, M449, and F540 labeled. The αC-helix is in yellow with E445 shown in sticks.

kinase domains (L390 in BTK) (*Figure 9a*). Regulatory domain release from the distal side of the kinase domain leads to disruption of the hydrophobic stack and is accompanied by exchange of ADP for ATP in the active site, thus linking the conformational preference of the regulatory domains and the occupancy of the active site (*von Raußendorf et al., 2017*). Given the impact of Ibrutinib and Dasatinib on the R-spine residue L460 and the fact that the hydrophobic stack residue Y461 is adjacent to L460, we can speculate that small molecules that fill (directly or indirectly) the back pocket of the active site, may allosterically trigger conformational adjustments in the hydrophobic stack region that result in displacement of the regulatory domains from their autoinhibited pose. This is a new metric for predicting global conformational effects of specific BTK inhibitors that does not depend on strict conformational requirements being met in the αC helix and/or activation loop.

Structural studies that combine data from X-ray crystallography, solution NMR and HDX-MS provide a detailed and more complete picture to identify key regulatory interactions at work in protein kinases. In addition, use of the full-length kinase reveals unexpected differences between BTK inhibitors that are missed in crystallographic analyses of fragments. For BTK, this work has demonstrated that the clinically important inhibitor Ibrutinib imposes conformational adjustments on the distant non-catalytic domains of BTK, which may be functionally relevant in vivo. BTK inhibitors that do not induce a conformational shift toward the 'open' form of BTK may improve the efficacy of disease treatment where the kinase-independent functions of BTK play an important role. Additionally, targeting downstream BTK substrates such as PLCγ2 or combination therapies of Ibrutinib together with reversible active site/allosteric inhibitors that have a prolonged serum half-life may reduce the type of resistance embodied by the T316A mutation. The sustained presence of reversible inhibitors in the serum may block the activity of newly synthesized BTK T316A and prevent escape from Ibrutinib. Recognizing the conformational impact that mutations and inhibitors such as Ibrutinib have on full-length BTK should inform future strategies to effectively target this kinase in the context of disease.

## Materials and methods

### Constructs and reagents

The bacterial expression constructs for murine BTK LKD, SH3-SH2-Linker-kinase domain (32LKD) and full-length (FL) have been described previously (*Joseph et al., 2017*). All BTK constructs carry the solubilizing Y617P mutation (*Joseph et al., 2017*). All mutations were made using the site-directed mutagenesis kit (Agilent), and the sequences of all constructs were confirmed by sequencing at the Iowa State University DNA facility. Ibrutinib, Dasatinib, CC-292 and CGI1746 were purchased from Selleckchem. GDC0853 was purchased from MedKoo Biosciences.

### Protein expression and purification

Full-length, kinase-active BTK, 32LKD, and BTK LKD were produced by co-expressing BTK with YopH in BL21(DE3) (Millipore Sigma) or BL21-Gold(DE3) cells (Agilent Technologies) as described previously (*Joseph et al., 2017*). Briefly, the culture was grown at 37°C to an O.D. 600 nm of 0.6 to 0.8. The temperature of the culture was lowered to 18°C and then induced with either 1.0 mM IPTG for BTK LKD construct or 0.1 mM IPTG for BTK full-length and 32LKD. The culture was harvested 24 hr after induction and the pellets were resuspended in lysis buffer (50 mM $KH_2PO_4$, pH 8.0, 150 mM NaCl, 20 mM imidazole and 0.5 mg/ml lysozyme) and stored at $-80$°C. Cells were lysed by thawing and the action of lysozyme, and 3000 U DNAse I (Sigma) and 1 mM PMSF were added to the lysate, incubated at RT for 20 min and then spun at 16,000 rpm for 1 hr at 4°C. Glycerol was added to the supernatant to a final concentration of 10% and was then incubated with Ni-NTA resin (QIAGEN) for 2 hr, washed with Tris pH 8.0, 75 mM NaCl, 40 mM imidazole, and eluted in 20 mM Tris pH 8.0, 150 mM NaCl, 250 mM Imidazole, and 10% glycerol. Eluted protein was flash frozen in liquid nitrogen

and stored at −80°C. The proteins were concentrated and dialyzed into the final NMR buffer which consists of 20 mM Tris pH 8.0, 150 mM Sodium chloride, 0.02% Sodium azide, and 10% glycerol. For HDX-MS analysis, the proteins were further purified by size exclusion chromatography (Hiload Superdex 26/60 200 pg or Hiload Superdex 26/60 75 pg, GE Healthcare). The fractions containing pure protein were pooled, concentrated, snap frozen and stored at −80° C. The final buffer consists of 20 mM Tris pH 8.0, 150 mM Sodium chloride, 0.02% Sodium azide and 10% glycerol. Phosphorylation level on Y551 of all purified BTK proteins used in this study is below western immuno-detection.

## NMR

Uniformly $^{15}$N-labeled BTK samples were produced as described earlier by growth in modified M9 minimal media containing $^{15}$N ammonium chloride (1 g/L, Cambridge Isotope Laboratories, Inc) as the sole source of nitrogen (*Joseph et al., 2017*). For the expression of BTK full-length, the M9 minimal media was supplemented with 0.1 mM Zinc chloride. The final NMR sample buffer consists of 20 mM Tris, 150 mM Sodium chloride, 10% glycerol, pH 8.0, and 0.02% Sodium azide. All NMR spectra were collected at 298 K on a Bruker AVIII HD 800 spectrometer equipped with a 5 mm HCN z-gradient cryoprobe operating at a $^{1}$H frequency of 800.41. NMR samples with inhibitors consisted of 150 μM $^{15}$N-labeled BTK, mixed with 200 μM inhibitor in 2% DMSO. All data were analyzed using NMRViewJ (*Johnson and Blevins, 1994*). Peak intensities were quantified using the integrate function within Bruker TopSpin.

## Intact mass spectrometry analysis of Ibrutinib-conjugated BTK

Prior to intact mass analysis, purified BTK full-length wild-type (20 μM) and Ibrutinib (40 μM), both in 20 mM Tris pH 8.0, 150 mM NaCl, 10% glycerol, 2% DMSO, were allowed to interact at 21°C for 1 hr. Both the free wild-type BTK, as a control, and Ibrutinib-labeled proteins (70 picomoles) were injected into a Waters nanoACQUITY with HDX technology set-up for intact mass analysis at 20°C. The proteins were injected into the sample loop and desalted for 3 min at 100 μL/min using water (0.1% formic acid) on an in-house packed POROS 20 R-2 trap. Proteins were eluted into the mass spectrometer using a 15–70% ACN (0.1% formic acid) gradient in 10 min at a flow rate of 100 μL/min. Mass spectra were acquired using a Waters Synapt HDMS$^{E}$ mass spectrometer operated in TOF only mode with a standard electrospray source, capillary voltage of 3200 V and a cone voltage of 40 V with a mass range of 50–2000 m/z. Intact mass values for both free and ibrutinib-labeled wild-type BTK were calculated from the raw m/z spectra using MaxEnt1 within MassLynx 4.1 (Waters) with a resolution of 0.10 Da and an output mass range of 65,000–90,000 Da.

## HDX-MS

General procedures for HDX-MS of BTK have been described in detail previously (*Joseph et al., 2017*). Details specific to experiments conducted here are provided in *Source data 1* in the format recommended (*Masson et al., 2019*) for HDX-MS experimental descriptions. All HDX-MS data have been deposited to the ProteomeXchange Consortium via the PRIDE (*Vizcaíno et al., 2016*) partner repository with the dataset identifier PXD020029. Briefly, prior to continuous labeling HDX experiments, purified BTK full-length wild-type, T316A, and C481S (20 μM) and Ibrutinib 40 μM, (20 mM Tris pH 8.0, 150 mM NaCl, 10% glycerol, 2% DMSO) were allowed to interact at 21°C for 1 hr (Note: Ibrutinib was only added to wild-type and T316A BTK). For the Ibrutinib titration experiment, labeling was performed for 10 min for BTK:Ibrutinib at molar ratios of 1:0.2, 1:0.5, 1:1, and 1:2. After the binding reactions, both the free kinase and kinase bound to Ibrutinib were placed on ice prior to deuterium labeling. Deuterium labeling proceeded for the times described using labeling buffer, and labeling was stopped with an equal volume of quench buffer at 0°C (details in *Source data 1*). Full-length wild-type BTK was independently mixed with CGI-1746, CC-292, GDC-0853, or Dasatinib to final concentrations of BTK 20 μM, inhibitor 200 μM and these mixtures were incubated for 1 hr at 21°C. Unbound kinase (as control) or kinase bound to each of the small molecules was deuterated for the times described using labeling buffer, and labeling was stopped with an equal volume of quench buffer at 0°C (details in *Source data 1*). Quenched samples were immediately analyzed using a Waters nanoACQUITY with HDX technology using online pepsin digestion with a Waters Enzymate immobilized pepsin column and UPLC separation of the resulting peptic peptides. Mass spectra were acquired using a Waters Synapt HDMS$^{E}$ mass spectrometer. Peptides generated from online

pepsin digestion were identified with Waters Protein Lynx Global Server 3.0 using separate unlabeled protein that was prepared in the same manner as protein labeled with deuterium. Deuterium incorporation was quantified using Waters DynamX 3.0. Deuterium levels for each peptide were calculated by subtracting the average mass of the undeuterated control sample from that of the deuterium-labeled sample; the data were not corrected for back exchange and are therefore reported as relative. Vertical difference maps in *Figures 5*, *7* and *8* do not represent a linear sequence of non-overlapping peptides. All coincident and overlapping peptides for comparisons in each figure are provided in figure identified tabs of *Source data 1*.

### Activity assays

In vitro kinase assays were performed by incubating 1 μM BTK FL, BTK FL C481S or BTK FL T316A proteins in a kinase assay buffer (50 mM Hepes pH 7.0, 10 mM $MgCl_2$, 1 mM DTT, 5% glycerol, 1 mM Pefabloc, and 200 μM ATP) at room temperature for 10 min. The reactions were stopped by the addition of SDS-PAGE loading buffer, and the samples were boiled, separated by SDS−PAGE, and western blotted with the anti-BTK pY551 antibody (BD Pharmingen) or anti-His antibody (EMD Millipore) as described previously (*Joseph et al., 2007*). The bands were quantified using the ChemiDoc (Biorad) gel imaging system. The phosphorylation levels (Anti-BTK pY551 blot) were normalized to the total protein level (Anti-His blot). The BTK FL value was set to one and compared to BTK FL C481S or BTK FL T316A. Initial phosphorylation levels of BTK WT, C481S, and T316A on Y551, prior to the start of the activity assay were undetectable by western immuno-detection.

## Acknowledgements

This work is supported by a grant from the National Institutes of Health (National Institute of Allergy and Infectious Diseases, AI43957) to AHA and JRE. The authors also thank the Roy J Carver Charitable Trust, Muscatine, Iowa for ongoing research support.

## Additional information

### Funding

| Funder | Grant reference number | Author |
| --- | --- | --- |
| National Institute of Allergy and Infectious Diseases | AI43957 | Amy Andreotti |

The funders had no role in study design, data collection and interpretation, or the decision to submit the work for publication.

### Author contributions

Raji E Joseph, Conceptualization, Formal analysis, Validation, Investigation, Writing - original draft, Writing - review and editing; Neha Amatya, D Bruce Fulton, Investigation; John R Engen, Data curation, Supervision, Funding acquisition, Project administration, Writing - review and editing; Thomas E Wales, Data curation, Validation, Investigation, Writing - review and editing; Amy Andreotti, Conceptualization, Formal analysis, Supervision, Funding acquisition, Writing - original draft, Project administration, Writing - review and editing

### Author ORCIDs

John R Engen  http://orcid.org/0000-0002-6918-9476
Amy Andreotti  https://orcid.org/0000-0002-6952-7244

### Decision letter and Author response

Decision letter https://doi.org/10.7554/eLife.60470.sa1
Author response https://doi.org/10.7554/eLife.60470.sa2

## Additional files

### Supplementary files
• Source data 1. Microsoft Excel file providing enhanced experimental details for HDX-MS (including minimum criteria specified by *Masson et al., 2019*), peptic peptide maps, lists of all peptides by residue number and sequence, and deuterium levels measured for each Figure. The value of each deuterium difference for every colored box in each Figure can be found in this file.

• Transparent reporting form

### Data availability
Hydrogen/deuterium exchange data have been deposited in the PRIDE database.

The following dataset was generated:

| Author(s) | Year | Dataset title | Dataset URL | Database and Identifier |
|---|---|---|---|---|
| Wales TE, Engen JR | 2020 | BTK regulatory domains escape Ibrutinib inhibition | https://www.ebi.ac.uk/pride/archive/projects/PXD020029 | PRIDE, PXD020029 |

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
