## [Decision Letter]

Thank you for submitting your article "BTK regulatory domains escape Ibrutinib inhibition" for consideration by *eLife*, along with a companion paper. Your two articles have been reviewed favorably by three peer reviewers, and the evaluation has been overseen by John Kuriyan as the Reviewing Editor and the Senior Editor. The following individual involved in review of your submission has agreed to reveal their identity: Ganesh Anand (Reviewer #2).

Despite the overall favorable response to both papers, the reviewers and the editors are in agreement that further consideration at *eLife* requires that both papers be merged into one (see reviews below). We are providing the same merged review for both papers. Note that only one resubmission link is provided for both papers.

The reviewers agree that the two papers (Joseph, Amatya et al "Differential impact of BTK active site inhibitors on the conformational state of full-length BTK"; Joseph, Wales, et al "BTK regulatory domains escape Ibrutinib inhibition") jointly represent substantial and important advances in the understanding of BTK, a clinically important kinase. Nevertheless, the reviewers have also identified some key weaknesses in each paper that would, on balance, preclude further consideration of the individual papers at *eLife*. The reviewers are strongly supportive of further consideration of a single merged manuscript. We recognize, however, that there may be important factors that prevent you from merging the two papers. Please let us know if that is the case – we will then release the two papers from *eLife*, allowing you to seek rapid publication of both papers elsewhere. On the other hand, support for a merged publication at *eLife* is strong, and we anticipate being able to give you a rapid decision after receiving a single merged resubmission.

Note that new experiments are not required in order to respond to the review – textual justifications in the merged manuscript will suffice. If you do have additional data that support points made in the revised manuscript we would, of course, welcome them.

Summary

The work by Joseph, Amatya et al. "Differential impact of BTK active site inhibitors on the conformational state of full-length BTK" examines the mechanism of inhibition of different Bruton's tyrosine kinase (BTK) active site inhibitor binding on the conformation of the protein.

This study addresses the allosteric effects of ATP-competitive inhibitors for Bruton's tyrosine kinase (BTK) and their mechanism. Using NMR and HDX solution measurements, the study compares one covalent inhibitor (ibrutinib) and four non-covalent inhibitors (GDC0853, CGI1746, CC292, dasatinib). The study reports three findings: First, NMR shows variable effects on W395 indole NH, which based on NMR and X-ray comparisons of linker-kinase BTK, reports active and inactive conformations of helix aC. These solution conformations differ from those predicted based on X-ray structures of each inhibitor bound to the BTK kinase domain. Second, HDX measurements show variable effects of inhibitors on FL-BTK. The results reveal clear conformational differences between BTK:drug complexes in solution, and an unexpected effect of dasatinib on increased conformational mobility that propagates throughout the kinase domain. Third, the binding interactions of each inhibitor with active site elements are used to construct a structural model which explains why only ibrutinib and dasatinib disrupt the remote SH3-SH2 interactions. The model proposes that L460 and/or T474 relay conformational changes from the active site to the regulatory domains through connections between R spine residues. This is then tested by mutation of L460A, which elevates HDX throughout the SH3-SH2 region as well as within the kinase domain, supporting its likely role in maintaining the allosteric connection between the active site and SH3-SH2 (but see concerns about effects on kinase stability, noted below). The T474A mutation does not affect the SH3-SH2 and kinase domain, but increases HDX in the linker region and autoinhibitory PHTH domain. Both mutants interfere with kinase activity partially (T474A) or fully (L460A).

The strengths of this manuscript are its analysis of the solution behavior of full length BTK using a powerful combination of NMR and HDX. The results establish evidence for the remote regulation of the SH3-SH2 regulatory domain, variable impact of different inhibitors on this allosteric mechanism, and variable conformational selection by different inhibitors within the active site. The model for allostery involving L460 as a sensor linking the kinase active site to the regulatory domains is exciting, and at least for ibrutinib is supported by HDX evidence. A principal weakness of the study is the limited rigorous mechanistic understanding of the effects of inhibitors on BTK conformation.

In Joseph, Wales, et al. "BTK regulatory domains escape Ibrutinib inhibition", the authors examine full length BTK using 1H-15N NMR and HDX-MS to investigate effects of an irreversible covalent inhibitor, ibrutinib, and a T316A drug resistance mutation whose mechanism is unknown. There are three main findings: First, NMR and HDX show that the inhibitor preserves the conformation of the kinase domain predicted from the X-ray structure, but disrupts remote autoinhibitory contacts between SH3-SH2 and the linker-kinase, leading to behavior resembling that of the partially active form. Second, the T316A resistance mutation enhances BTK kinase activity, while increasing HDX in the SH3-SH2 domains, consistent with disengagement to a partially activated state, and shifting a W395 NH resonance towards a productive active site conformation. Third, the effect of ibrutinib on remote conformational changes is nearly the same between WT-BTK and the T316A mutant. The study concludes that ibrutinib has long distance allosteric effects, displacing the SH3 and SH2 domains from their compact autoinhibited conformation and that T316A resistance mutation located outside of the kinase domain works by increasing the catalytic activity of BTK.

Strengths of the manuscript are its unique combination of NMR and HDX, and its excellent illustration of the power of this approach in probing regulatory conformational changes, in a way that has been elusive using X-ray crystallography. The description of NMR assignments are rigorous and the data interpretation, particularly with HDX are accurate and not overstated. There are few studies of full-length BTK, and together with the co-submitted article, the manuscript makes a clear case for allostery unique to the full-length form that can be regulated by a clinically important inhibitor and a poorly understood resistance mechanism. Weaknesses are that the manuscript is confusing to understand without extensive reading of previous articles in Structure 2017 and JMB 2016. It would benefit from revisions that clarify a number of ambiguities and confusing terminologies. In addition, the study needs a more convincing explanation for why the results necessarily explain drug resistance by T316A. HDX-MS experiments indicate that despite stabilizing an inactive conformation of the kinase domain, ibrutinib induces disassembly of the inhibitory interactions with the SH3 and SH2 domains. This is a surprising finding that could have important implications for the effects of the drug, which is an FDA-approved therapeutic. Studies with the T316 mutant show that it induces disassembly of the SH3/SH2 interactions and consequently enhances BTK catalytic activity, but the molecular explanation for this is unsatisfactory.

Comments to address when revising and merging the manuscripts:

A) Specific comments pertaining to Joseph, Amatya, et al. "Differential Effects of BTK active site inhibitors.…"

Major points to address:

1) There is concern that the L460 and T474 mutations are hard to interpret – they are non-conservative substitutions in key regions of the kinase domain that almost certainly destabilize it to some extent, and can therefore be expected to affect SH3/SH2 regulatory interactions. Indeed the authors see this for the L460A mutation. So it may be difficult to ascribe HDX changes with inhibitors to a role for these residues in allosteric communication. It is possible that similar results with many non-conservative substitutions in this region of the kinase domain. The authors should consider either removing these experiments, or justify their inclusion with additional text.

2) "The BTK:CC292 complex shares structural similarity with the ibrutinib, GDC and CGI BTK complex crystal structures but the solution behavior of the CC292 complex is distinct. HDX reveals only limited protection upon CC292 binding localized primarily to the N-lobe surrounding the active site."

Detailed analyses of drug-enzyme interactions are needed to support this statement. Superficial examination of Figure 1A suggests that the BTK:CC292 cocrystal structure might have more space for water in the active site.

Related to this point, it would be helpful to note in the manuscript which of these structures were generated from co-crystallized proteins and which were generated by soaking.

3) T474A does not phenocopy effects of L460A on HDX (see also the comment above on the potential effects of mutations on stability). Therefore, it was hard to understand how the proposed mechanism explains why dasatinib necessarily phenocopies ibrutinib effects on HDX within the remote SH3-SH2 regions. The conclusions about T474A are not integrated very well with results in Figures 3 and 4 to explain the distinct effects of dasatinib. The authors propose that both ibrutinib and dasatinib perturb L460 in different ways, where ibrutinib interacts directly with L460 to push helix aC out, while dasatinib interacts with T474 to allow helix aC-in, which also perturbs the back pocket. The part of the mechanism involving dasatinib is largely based on the X-ray structure, which seems incongruent with the conclusion from Figure 3 that the Xray structures for CC292 and dasatinib are not representative of their solution conformations. Therefore, it seems possible that perturbation of T474 (the gatekeeper) signals to the remote SH3-SH2 by an alternative route. Perhaps prior findings in JMB 2016 showing that T474M activates the kinase would be useful to integrate into this Discussion. It might also be useful to show the outcomes of Figure 8 experiments using dasatinib instead of ibrutinib.

Overall these two large sections read somewhat incoherently, and it was difficult to parse out which aspects were supported by evidence and which were speculation. The sections need to be reconsidered or explained more carefully.

4) The authors should place their model in the context of “Conformational Selection” , which is a general mechanism that could explain all their results.

B) Specific comments pertaining to Joseph, Wales, et al. (BTK regulatory domains escape.…)

Major points to address:

1) It would help to clarify that the active mutants used in the prior work as well as T316A in this study only activate BTK by 2-3x. Therefore all mutants cause only a partially active state of BTK, that is intermediate between the autoinhibited and fully active kinase. This was hard to grasp from the manuscript, but explained in a short review of the 2017 Structure paper by Agnew and Jura. Thus the current manuscript is somewhat misleading in using "active conformation" terminology throughout. Is there work comparing HDX in full length BTK vs SH3 and SH2 domains alone? It is possible that the HDX increase would be much greater in fully activated BTK than in the 2-fold activated mutants. It would help to say this, and explain that the mutants increase HDX by only 1-2 Da because the detachment reflects only a partially activated intermediate conformation.

Related to this, can any form of FL-BTK yield full activation in vitro? For example, does stoichiometric phosphorylation at Y551 by Lyn and/or autophosphorylation at Y223, with or without PIP3 binding, result in a fully active kinase? If available, it would be useful to know how HDX of SH3-SH2 in the fully activated state compares to that of the mutants.

2) Given prior data, including the crystal structure of the BTK kinase domain with ibrutinib and crystal structure of a SH3/Sh2/kinase fragment of BTK in the autoinhibited state, it might be expected that ibrutinib would tend to stabilize the assembled, autoinhibited state of BTK because it stabilizes the inactive conformation of the kinase domain (which is in turn coupled to SH3/SH2 interactions). But the present HDX-MS data clearly suggests otherwise. Is it possible that the destabilization arises from binding of ibrutinib to a solvent exposed cysteine elsewhere in BTK – for example the one in the SH2 domain? The authors present beautiful mass spec data showing 1:1 labeling with ibrutinib, but was this labeling experiment carried out under the same conditions as ibrutinib binding for HDX-MS? Also the mass spec experiment shows not even a trace of unlabeled BTK or double labeled BTK – was some sort of mass selection or filtering applied in this experiment to select the singly-labeled species? Also, Materials and methods should be provided for this mass spec experiment.

3) It was unconvincing that T316A should promote resistance by activating FL-BTK.

First, the extent of activation by the mutation was only 2-fold. The assay measures autophosphorylation at Y551, which only occurs in vitro and might be irrelevant to cellular activation, therefore might fail to report potentially larger effects of the mutation on activity measured with a true substrate or autophosphorylation at pY223.

Related to point #3 above, what is the effect of the T316A mutation on the activity of a fully activated BTK (if it exists), and the degree of its inhibition by ibrutinib? Perhaps T316A increases activity to a greater extent when BTK is fully phosphorylated.

Second, as noted by the authors, the irreversible coupling of ibrutinib might mask any effects of the mutant on binding affinity. The latter might be detectable using BTK-C481A to measure ibrutinib binding, or by measuring rates of covalent coupling of ibrutinib to BTK at varying concentrations of a competing inhibitor.

Third, elevated activity of unbound BTK-T316A would not be expected to bypass the effects of drug, if drug truly binds the mutant in cells.

4) Enzymes exist in ensembles of multiple conformations in solution. This is presented well with the TROSY 1H15N spectra in Figure 2. Crystallography provides high resolution snapshots of only one likely conformation. It is erroneous for the authors to validate their ensemble measurements with crystal structures. The conclusions from the NMR titrations in Figure 2A are correlated with inactive state in Figure 1D. If anything, the authors' results caution against relying on simplistic single snapshot conformational analyses to explain allostery.

5) It will greatly help the reader if the authors discuss and reassess their results on the basis of “Conformational Selection”, “Induced Fit” or a combination of both models. It could be argued that the inhibitor favors one conformation and shifts the equilibrium. The authors throughout the manuscript imply Ibrutinib binding allosterically displaces the remote regulatory domain- and then there is a notion of regulatory domains escaping the inhibitor from the title.

Supporting the ensemble behavior and conformational selection, the authors conclude from the NMR and HDXMS studies that the T316A mutant shifts the ensemble to a more active kinase conformation consistent with a mutation-induced bias toward the active state.

6) An important control would be to compare HDXMS upon Ibrutinib binding with a non-specific covalent modification of BTK at C481 with biotin maleimide for instance. An alternate control would be analysis of the C481M mutant, which might mimic the covalent modification of C481. Have such experiments been done?

In this context, the noncovalent interaction of Ibrutinib with C481S is intriguing. Does it give the same equivalent inhibitory effect? Does it compete equivalently with ATP? Are the HDX-MS differences equivalent to wt C481?

[Editors' note: further revisions were suggested prior to acceptance, as described below.]

Thank you for submitting your revised article "Differential impact of BTK active site inhibitors on the conformational state of full-length BTK" for consideration by *eLife*. Your article has been reviewed by three peer reviewers, and the evaluation has been overseen by a Reviewing Editor and John Kuriyan as the Senior Editor. The following individuals involved in review of your submission have agreed to reveal their identity: Natalie G Ahn (Reviewer #1); Ganesh Srinivasan Anand (Reviewer #2); Michael J Eck (Reviewer #3).

Your paper is now suitable for publication in *eLife*. Note that one of the reviewers has raised some important points of clarification. Please edit the manuscript to incorporate your response to these comments. We leave this to your discretion, and the revised manuscript will be accepted for publication immediately, without further editorial intervention.

Reviewer #1:

The revised manuscript addresses all of my concerns raised in the initial review and is greatly improved in coherency and readability. The current manuscript represents a valuable contribution that illustrates the power of combined NMR and HDX-MS approaches to understand solution conformational changes caused by ligand binding to and mutations of protein kinases that are clinically targeted.

Reviewer #2:

The revised manuscript is much improved and I support publication of the work.

The following points should be addressed by the reviewers:

1) In the Abstract the authors state "including unexpected shifts in the

conformational equilibria of the regulatory domains." Allosteric propagation from catalytic to regulatory domains are more the norm than the exception. The authors should consider replacing “unexpected” with “propagated” or large-scale etc.

2) The authors refer to the comparative magnitude protection in deuterium exchange for one drug molecule compared to another as “more intense”. Protection is either more or less! “Intense” is better suited to describing function.

3) I would like to point the authors to a recent publication that addresses integrative allostery by two drugs on kinases by Ghode et al., 2020, Biophysical Journal "Synergistic allostery in multiligand-protein interactions"

Reviewer #3:

The combined manuscript is much improved and the authors have addressed my concerns. I support publication of the revised work in *eLife*.

---

## [Author Response]

Comments to address when revising and merging the manuscripts:A) Specific comments pertaining to Joseph, Amatya, et al. "Differential Effects of BTK active site inhibitors.…"Major points to address:1) There is concern that the L460 and T474 mutations are hard to interpret – they are non-conservative substitutions in key regions of the kinase domain that almost certainly destabilize it to some extent, and can therefore be expected to affect SH3/SH2 regulatory interactions. Indeed the authors see this for the L460A mutation. So it may be difficult to ascribe HDX changes with inhibitors to a role for these residues in allosteric communication. It is possible that similar results with many non-conservative substitutions in this region of the kinase domain. The authors should consider either removing these experiments, or justify their inclusion with additional text.

As per the reviewer’s suggestion, experiments involving BTK L460 and T474 have been removed from the merged manuscript. The potential role of L460 is now described exclusively in the Discussion section.

2) "The BTK:CC292 complex shares structural similarity with the ibrutinib, GDC and CGI BTK complex crystal structures but the solution behavior of the CC292 compex is distinct. HDX reveals only limited protection upon CC292 binding localized primarily to the N-lobe surrounding the active site."Detailed analyses of drug-enzyme interactions are needed to support this statement. Superficial examination of Figure 1A suggests that the BTK:CC292 cocrystal structure might have more space for water in the active site.

This section of the manuscript has been edited and expanded to discuss the occupancy of CC-292 within the active site as suggested.

Related to this point, it would be helpful to note in the manuscript which of these structures were generated from co-crystallized proteins and which were generated by soaking.

The figure caption (Figure 3) has been modified to address this point.

3) T474A does not phenocopy effects of L460A on HDX (see also the comment above on the potential effects of mutations on stability). Therefore, it was hard to understand how the proposed mechanism explains why dasatinib necessarily phenocopies ibrutinib effects on HDX within the remote SH3-SH2 regions. The conclusions about T474A are not integrated very well with results in Figures 3 and 4 to explain the distinct effects of dasatinib. The authors propose that both ibrutinib and dasatinib perturb L460 in different ways, where ibrutinib interacts directly with L460 to push helix aC out, while dasatinib interacts with T474 to allow helix aC-in, which also perturbs the back pocket. The part of the mechanism involving dasatinib is largely based on the X-ray structure, which seems incongruent with the conclusion from Figure 3 that the Xray structures for CC292 and dasatinib are not representative of their solution conformations. Therefore, it seems possible that perturbation of T474 (the gatekeeper) signals to the remote SH3-SH2 by an alternative route. Perhaps prior findings in JMB 2016 showing that T474M activates the kinase would be useful to integrate into this Discussion. It might also be useful to show the outcomes of Figure 8 experiments using dasatinib instead of ibrutinib.Overall these two large sections read somewhat incoherently, and it was difficult to parse out which aspects were supported by evidence and which were speculation. The sections need to be reconsidered or explained more carefully.

As per the reviewer’s suggestion earlier (in point #1), experiments involving BTK L460 and T474 have been removed from the merged manuscript. Instead we now discuss a potential role for the back pocket residue L460 (part of the regulatory spine) in the Discussion section.

4) The authors should place their model in the context of “Conformational Selection” , which is a general mechanism that could explain all their results.

We agree that conformational selection is an appealing mechanism to explain the observations in this work. However, unlike the excellent study published very recently on ABL (Xie et al., 2020), we have not characterized the distinct conformations populated by full-length apo BTK and so it is not possible to unequivocally ascribe the changes we observe upon drug binding or mutation to a conformational selection model, induced fit model, or a combination of both.

B) Specific comments pertaining to Joseph, Wales, et al. (BTK regulatory domains escape.…)Major points to address:1) It would help to clarify that the active mutants used in the prior work as well as T316A in this study only activate BTK by 2-3x. Therefore all mutants cause only a partially active state of BTK, that is intermediate between the autoinhibited and fully active kinase. This was hard to grasp from the manuscript, but explained in a short review of the 2017 Structure paper by Agnew and Jura. Thus the current manuscript is somewhat misleading in using "active conformation" terminology throughout. Is there work comparing HDX in full length BTK vs SH3 and SH2 domains alone? It is possible that the HDX increase would be much greater in fully activated BTK than in the 2-fold activated mutants. It would help to say this, and explain that the mutants increase HDX by only 1-2 Da because the detachment reflects only a partially activated intermediate conformation.

This is an excellent point. We have previously compared isolated SH3 and SH2 domains to full-length BTK by HDX-MS and indeed the magnitude of the increase in deuterium uptake is small in the full length T316A mutant when compared to either 1) the isolated domains, 2) the D656K activating mutation or perhaps more relevant, 3) binding of full-length BTK to PIP3, an activating ligand that binds the PH domain (all three cases were described in Joseph, Structure 2017). This suggests that the T316A mutation induces a small shift in the conformational ensemble of the regulatory domains of full-length BTK away from the autoinhibited conformation (as compared to the larger shift promoted by PIP3 binding). This is now carefully acknowledged in the text describing the effect of T316A.

The manuscript has now been modified throughout to clearly indicate the conformational state probed by the NMR experiment. The “active” versus “inactive” state terminology has been replaced in the NMR experiments to reflect the αC-in versus -out conformation of the kinase domain. A panel has been added to Figure 2 to more clearly show the αC helix and W395 conformations in structures of the active and inactive kinase. The HDX-MS changes are interpreted in the context of the autoinhibited structure since this is the only available multidomain structure for BTK. The data show the location of changes that are consistent with release of the regulatory domains from the kinase domain.

Related to this, can any form of FL-BTK yield full activation in vitro? For example, does stoichiometric phosphorylation at Y551 by Lyn and/or autophosphorylation at Y223, with or without PIP3 binding, result in a fully active kinase? If available, it would be useful to know how HDX of SH3-SH2 in the fully activated state compares to that of the mutants.

The PIP3 bound, Y551 phosphorylated full-length BTK protein is presumed to be the fully activated state, however we have not characterized this form due to challenges associated with producing sufficient uniformly Y551 phosphorylated sample. For this reason, all NMR and HDX-MS experiments have been carried out using unphosphorylated BTK to ensure sample homogeneity.

2) Given prior data, including the crystal structure of the BTK kinase domain with ibrutinib and crystal structure of a SH3/Sh2/kinase fragment of BTK in the autoinhibited state, it might be expected that ibrutinib would tend to stabilize the assembled, autoinhibited state of BTK because it stabilizes the inactive conformation of the kinase domain (which is in turn coupled to SH3/SH2 interactions). But the present HDX-MS data clearly suggests otherwise. Is it possible that the destabilization arises from binding of ibrutinib to a solvent exposed cysteine elsewhere in BTK – for example the one in the SH2 domain? The authors present beautiful mass spec data showing 1:1 labeling with ibrutinib, but was this labeling experiment carried out under the same conditions as ibrutinib binding for HDX-MS? Also the mass spec experiment shows not even a trace of unlabeled BTK or double labeled BTK – was some sort of mass selection or filtering applied in this experiment to select the singly-labeled species? Also, Materials and methods should be provided for this mass spec experiment.

In the HDX MS experiments, we were not able to identify any cysteine-containing peptides that were conjugated to Ibrutinib, including those peptides containing C481. However, peptic peptides that contain C481 were absent only in the Ibrutinib labeled BTK samples while all other C containing peptides retained signal in the mass spectrometer suggesting that Ibrutinib conjugated only to C481. We are therefore confident, when combined with intact MS analysis (Figure 2C), that there were no substantial sideconjugation reactions to other cysteines in the protein.

Thank you for pointing out our error in not describing the methods for the acquisition of the intact mass data. The complete methods for the intact mass analyses are now included in the Materials and methods section of the text. These added methods describe that intact mass analysis was performed under the same conditions as those for the HDX MS experiments, and during mass spectral acquisition there was no mass selection or filtering applied for either the Ibrutinib conjugated or free wild-type BTK. Mass spectra were also acquired after 4 hours of labeling BTK with Ibrutinib and still there was a single mass adduct (data not shown).

3) It was unconvincing that T316A should promote resistance by activating FL-BTK.First, the extent of activation by the mutation was only 2-fold. The assay measures autophosphorylation at Y551, which only occurs in vitro and might be irrelevant to cellular activation, therefore might fail to report potentially larger effects of the mutation on activity measured with a true substrate or autophosphorylation at pY223.Related to point #3 above, what is the effect of the T316A mutation on the activity of a fully activated BTK (if it exists), and the degree of its inhibition by ibrutinib? Perhaps T316A increases activity to a greater extent when BTK is fully phosphorylated.

We acknowledge that the kinase assay measuring phosphorylation on Y551 might occur only in vitro. However, this assay does report on the intrinsic activity of the kinase and shows a 2-fold increase in activity which is consistent with the population shift measured by NMR. The drug resistance mutation in the ABL SH2 domain, T231R, has been studied extensively and, while this substitution directly stabilizes the active ABL conformation, the fold activation observed using an in vitro kinase assay is also two-fold (Sherbenou et al., 2010). In this case drug resistance is also thought to arise from a conformational shift away from a form to which the drug binds. While the T316A mutation could have additional effects in the context of a fully phosphorylated form of BTK, the impact of the mutation on the unphosphorylated protein is likely playing a role in Ibrutinib resistance in vivo given the pharmacokinetics of this drug. Finally, we are aware of data (personal communication) supporting a resistance causing mutation in a SRC module kinase that does not affect drug binding to the active site.

Second, as noted by the authors, the irreversible coupling of ibrutinib might mask any effects of the mutant on binding affinity. The latter might be detectable using BTK-C481A to measure ibrutinib binding, or by measuring rates of covalent coupling of ibrutinib to BTK at varying concentrations of a competing inhibitor.

We did consider these approaches to measure binding affinity. Using the C481A mutant to measure binding affinity of Ibrutinib would be an approximation and not reflect the true binding kinetics of a covalent inhibitor. The use of a competing inhibitor (eg. fluorescent tagged ADP) would require that it binds with the same affinity to both the wild-type and mutant proteins. However, previous work has shown that destabilization of the autoinhibited conformation changes the affinity for ADP. Given these complications, we tested binding using HDX-MS and acknowledge the deficiencies in the approach.

Third, elevated activity of unbound BTK-T316A would not be expected to bypass the effects of drug, if drug truly binds the mutant in cells.

The pharmacokinetics of an inhibitor is a key factor that must be taken into consideration in assessing mutational outcomes. Covalent inhibitors such as Ibrutinib, unlike reversible inhibitors, are rapidly cleared from the serum and Ibrutinib levels drop to basal levels within 4 hours. Newly synthesized BTK T316A could escape this narrow inhibition time window. The unconjugated T316A mutant therefore could promote BTK signaling due to its elevated activity, unlike wild-type BTK which is autoinhibited. In the merged manuscript we address this at the beginning of the Discussion section. In addition, we have acknowledged in this section that the T316A mutation could also adversely affect Ibrutinib binding to the active site thereby contributing to resistance especially in the context of activation loop phosphorylation.

4) Enzymes exist in ensembles of multiple conformations in solution. This is presented well with the TROSY 1H15N spectra in Figure 2. Crystallography provides high resolution snapshots of only one likely conformation. It is erroneous for the authors to validate their ensemble measurements with crystal structures. The conclusions from the NMR titrations in Figure 2A are correlated with inactive state in Figure 1D. If anything, the authors' results caution against relying on simplistic single snapshot conformational analyses to explain allostery.

We agree that our solution work are ensemble measurements and that crystal structures provide snapshots of a single conformation. Inhibitor and mutational effects are interpreted as shifts in this conformational equilibrium and the crystal structures are used as a guide to interpret these changes.

5) It will greatly help the reader if the authors discuss and reassess their results on the basis of “Conformational Selection”, “Induced Fit” or a combination of both models. It could be argued that the inhibitor favors one conformation and shifts the equilibrium. The authors throughout the manuscript imply Ibrutinib binding allosterically displaces the remote regulatory domain- and then there is a notion of regulatory domains escaping the inhibitor from the title.Supporting the ensemble behavior and conformational selection, the authors conclude from the NMR and HDXMS studies that the T316A mutant shifts the ensemble to a more active kinase conformation consistent with a mutation-induced bias toward the active state.

It is noteworthy that the recent publication referenced above (Xie et al., 2020) is one of the few examples in the kinase field where multiple structures within an ensemble have been detected and structures solved so that conformational shifts induced by mutations could be correlated to structures that exist in the ensemble of the wild type protein – supporting conformational selection. The NMR approach used in these studies (CEST) requires long acquisition times that might prohibit application to full-length BTK.

6) An important control would be to compare HDXMS upon Ibrutinib binding with a non-specific covalent modification of BTK at C481 with biotin maleimide for instance. An alternate control would be analysis of the C481M mutant, which might mimic the covalent modification of C481. Have such experiments been done?In this context, the noncovalent interaction of Ibrutinib with C481S is intriguing. Does it give the same equivalent inhibitory effect? Does it compete equivalently with ATP? Are the HDX-MS differences equivalent to wt C481?

Ibrutinib and CC-292 (also in this study) are both covalent inhibitors that modify BTK C481. CC292 can therefore be considered as a control for Ibrutinib and in the general sense modification at C481 – the consequences of Ibrutinib and CC-292 on the overall conformational preferences and dynamics of full-length BTK are strikingly different.

C481S is an Ibrutinib resistance mutation. The C481S mutant exhibits an increase in IC_50_ for Ibrutinib in vivo. ATP binding to the BTK C481S mutant has not been tested. Our NMR and HDX-MS analysis of the C481S mutant and wild-type protein show no significant differences between the apo proteins (Figure 7 and 8 in the manuscript) or when bound to Ibrutinib (data not shown). It is likely that loss of the covalent bond to Ibrutinib results in a faster off rate and subsequent clearance of Ibrutinib from the serum, in other words BTK C481S is not “trapped” by the drug.